# High performance mechano-optoelectronic molecular switch

Zhenyu Yang[1,5], Pierre-André Cazade[2,5], Jin-Liang Lin[1,5], Zhou Cao[1], Ningyue Chen[1], Dongdong Zhang [1,3], Lian Duan [1,3], Christian A. Nijhuis [4], Damien Thompson [2] ✉ & Yuan Li [1] ✉

Highly-efficient molecular photoswitching occurs ex-situ but not to-date inside electronic devices due to quenching of excited states by background interactions. Here we achieve fully reversible in-situ mechano-optoelectronic switching in self-assembled monolayers (SAMs) of tetraphenylethylene molecules by bending their supporting electrodes to maximize aggregation-induced emission (AIE). We obtain stable, reversible switching across >1600 on/off cycles with large on/off ratio of $(3.8 \pm 0.1) \times 10^3$ and $140 \pm 10$ ms switching time which is 10-100× faster than other approaches. Multimodal characterization shows mechanically-controlled emission with UV-light enhancing the Coulomb interaction between the electrons and holes resulting in giant enhancement of molecular conductance. The best mechano-optoelectronic switching occurs in the most concave architecture that reduces ambient single-molecule conformational entropy creating artificially-tightened supramolecular assemblies. The performance can be further improved to achieve ultra-high switching ratio on the order of $10^5$ using tetraphenylethylene derivatives with more AIE-active sites. Our results promise new applications from optimized interplay between mechanical force and optics in soft electronics.

Simplified and energy-efficient electronic devices that respond to multiple external stimuli (*e.g.*, voltage, light, and mechanical stress) are needed for nascent technologies ranging from soft robotics and neuromorphic computing to Internet-of-Things[1–3]. Yet most research to date focused on one switching modality with one stimulus[4–8]. Here we align materials design with device technology by introducing mechanical control over photoswitching leading to a new type of dual-gated molecular switch. While molecular switches are inherently energy-efficient[9], theoretically ultrafast molecular photoswitches have shown disappointing performance to date[6–8]. Therefore, there is an increasing need to move beyond ex-situ switching[8] and develop efficient in-situ molecular switches controlled by light, which has proved challenging due to quenching and spontaneous back-switching[6, 7]. On the other hand, molecular mechanical switches have been seldom reported[10], despite wide implementation of mechanically controlled switches[11–13] and recent report of miniaturized opto-electro-mechanical switching using a 25 nm gold membrane[14]. Furthermore, the range of applications in fields such as circularly polarized luminescence (CPL) has been expanded using stimuli of temperature, humidity, and pH[15]. While ex-situ photoswitching has found some applications, such as in gold nanocluster-based fluorescence photoswitching[16], further advances require in-situ and multi-stimuli

[1]Key Laboratory of Organic Optoelectronics, Department of Chemistry, Tsinghua University, Beijing 100084, P.R. China. [2]Department of Physics, Bernal Institute, University of Limerick, Limerick V94 T9PX, Ireland. [3]Laboratory of Flexible Electronics Technology, Tsinghua University, Beijing, P.R. China. [4]Department of Molecules and Materials MESA+ Institute for Nanotechnology, Molecules Center and Center for Brain-Inspired NanoSystems Faculty of Science and Technology, University of Twente, Enschede, The Netherlands. [5]These authors contributed equally: Zhenyu Yang, Pierre-André Cazade, Jin-Liang Lin. ✉e-mail: damien.thompson@ul.ie; yuanli_thu@tsinghua.edu.cn

switching techniques. Here, we use mechanical bending of the supporting electrode to direct molecular self-assembly of aggregation-induced emission (AIE) active tetraphenylethylene molecules[17,18] which allows us modulate the current under both light and mechanical force. This results in rapid, strong, reliable and sustained molecular switching. The robust high-performance photoswitch is 10–100 times faster than other approaches with on/off ratio of $(3.8 \pm 0.1) \times 10^3$ during 1600 bright/dark cycles under mechanical force, providing an alternative design route for flexible electronics and optomechatronics. Ultra-high photoswitching on the order of $10^5$ is achieved using tetraphenylethylene derivatives with more AIE-active sites, showing that the device performance can be rationally engineered at the molecular level.

To reach large on/off ratios, molecular photoswitches[6] rely on optically triggered conformational changes such as photoisomerization[19,20] or photocyclization[21–23]. However, reported molecular photoswitches with a high on/off ratio ($>100$), even reaching $10^4$ recently[22,24], need a long illumination time ($>30$ min) to guarantee sufficient isomerization for switching conductivity. Those photoswitching processes are non-volatile and have promising molecular device applications in information storage and processing[6,8,12,15]. However, these non-volatile photoswitches cannot provide the in-situ fast switching required by memory devices in electrical circuits[12,19]. In contrast to the photoisomerization-based approach, photo-assisted conduction via photostationary states is a volatile switch. For example, fast molecular photoswitches with switching speeds of <1 s have been reported for systems that avoid conformational changes, using instead shifts of molecular orbital energy levels that enhance tunneling in bright. Yet these approaches still suffer from low on/off ratios (~1.4) and low stability[25]. To date, the best photoswitches were demonstrated by the landmark report from Guo et al.[8] of a graphene–diarylethene–graphene single-molecule junction with on/off ratio of more than $10^2$ and switching time of less than 100 ms. Chiechi et. al.[26] showed large-area self-assembled monolayers (SAMs) of hemicyanine molecules generating a similar on/off ratio of ~100 with switching time of less than 60 s. The challenge then to achieve simultaneous large on/off ratio, fast, stable, reversible in-situ switching in one molecular device requires the generation of stable on and off states that do not suffer from unwanted side-effects of photo-quenching or thermal relaxation, and in addition, do not undergo light-induced conformational changes. We hypothesized that by incorporating AIE into flexible molecular tunnel junctions, we could bridge the photo- and mechano-responses of organic molecules[14,27,28] to create the first mechano-optoelectronic molecular switch. In this work, we demonstrate fast, large, reversible and sustained optically controlled current switching, enabled by mechanically controlled molecular assembly. The programmable molecular switches (Fig. 1a) are SAMs derived from HS(CH$_2$)$_{10}$-O-tetraphenylethylene molecules (abbreviated as HSC$_{10}$-O-TPE; the synthesis and characterization of the molecules are shown in Supplementary Figs. 1–13) on a 30 nm thick ultra-flat and semi-transparent gold (Au) substrate attached to a flexible and transparent polyethylene terephthalate (PET) film. Figure 1b, c illustrates the use of the mechanically controlled bending curvature at the millimeter scale to direct the supramolecular assembly at the nanometer scale. A top-electrode of eutectic indium-gallium (EGaIn) liquid-metal alloy was used to create electrical junctions of the form PET/Au-SC$_{10}$-O-TPE//Ga$_2$O$_3$/EGaIn, and a UV lamp (365 nm) was focused on the bottom of the junctions for light-controlled switching. The extent of the bending geometry was captured on camera and we extracted the bending radius ($R$) from the substrate curvature (Supplementary Figs. 59–61). We observed rapid and robust light-controlled switching of current density ($J$, A/cm$^2$) with an average on/off ratio (defined as $J_{UV\text{-}on}/J_{UV\text{-}off}$ at -1.0 V) of $(3.8 \pm 0.1) \times 10^3$ obtained from a large dataset of over 1600 reversible cycles (Fig. 1d; see Supplementary Figs. 62 and 63 for additional data sets) with very

short $140 \pm 10$ ms switching time (Fig. 1e), when the SAMs of PET/Au-SC$_{10}$-O-TPE were concavely bent to a radius of $R = 17.0$ mm under illumination. Figure 1f shows the overall average on/off ratio as a function of all the measured $R$ values, demonstrating precise mechanical control of the molecular photoswitching, and showing also their potential as electrical mechanosensors. This capability allows us to control the degree of aggregation of the photo-active TPE groups by scaling the strength of their supramolecular packing interactions, which enables reliable and sustained high-performance light-controlled molecular switching.

## Results

### Mechano-optoelectronic molecular switch

The mechanism behind the mechanically controlled photoswitching is schematically illustrated in Fig. 1g, h. The simplified Jablonski diagram in the inset of Fig. 1g illustrates how the fluorescent emission of the TPE molecules, when they are not aggregated, is minimal due to vibrational relaxation from the rotations of the four phenyl rings. In this case, the extent of current flow through the junctions in the flat geometry is limited by the large energy offset ($\delta E_{ME} = 1.6$ eV, calculated from the experimental data of UPS spectrum) between the highest occupied molecular orbital (HOMO) energy level of the TPE moieties and the Fermi level of the electrodes (Fig. 1g), and the charge transport through those junctions cannot be photoswitched. When the SAMs are concavely bent inducing TPE aggregation (Fig. 1h), the vibrational relaxations are largely suppressed with a strong photoexcitation of the TPE molecules, from which we observe a large increase of current indicating a lowering of the tunneling barrier, which reduces $\delta E_{ME}$. Hence the high-performance switching of conductivity by light on/off can be achieved in the concavely bent device geometry.

To demonstrate a robust photoswitch, we measured statistically large numbers of log$|J|$($V$) curves and constructed Gaussian log-average values of $J$ (log$|J|$) vs. $V$ curves following previously reported methods (see Supplementary Figs. 66–68 for details including all histograms and fits of $J$ at -1.0 V). The optical modulation of log$|J|$ across the SAMs of Au-SC$_{10}$-O-TPE can be decreased to unity when the junctions are at maximally convex $R = -18.3$ mm (Fig. 2a) and show a small on/off ratio of 7.0 when the junctions are flat (Fig. 2b). By contrast, we measure a very large difference in log$|J|$ with an average on/off ratio of $3.8 \times 10^3$ when the junctions are at the maximally concave bending of $R = 17.0$ mm (Fig. 2c). The control junctions with photo-inactive SAMs that lack the TPE unit, Au-SC$_{10}$ (Fig. 2d) and Au-SC$_{18}$ (Fig. 2e), show no difference in log$|J|$($V$) with light stimulus as a function of concave or convex bending (Supplementary Fig. 68). Therefore, we rule out the thermal effects of UV illumination or possible plasmon resonance-induced tunneling as the source for the large increase in log$|J|$($V$) by UV light (see ref. 29 for discussion of light-induced effects in junctions). These experiments prove that photoswitching is a molecular effect and indicate that the current switching depends on the mechanically modulated aggregation of the TPE headgroups. Figure 2f shows temperature-dependent log$|J|$($V$) measurements with the light on and off at temperatures of 40, 60, and 80 °C. We find statistically indistinguishable log$|J|$($V$) curves at elevated and ambient temperature (25 °C, Fig. 2c), confirming temperature-independent coherent tunneling as the dominant mechanism of charge transport in both the UV-on and UV-off states. The much larger value of $\delta E_{ME}$ (1.6 eV) compared to the $\pm 1.0$ eV of the chemical potential applied at the Fermi level of the electrodes means that the current at UV-off can be explained by an off-resonance tunneling model. The transition voltage analysis shown in Supplementary Fig. 71 suggests the conduction model remains unchanged between UV-on and UV-off[25,30]. Therefore, we conclude that the enhancement of current under illumination is driven by the reduction of the tunneling barrier, likely caused by the creation of an in-gap SOMO level (singly occupied molecular orbital) of the photo-excited molecular cation[26], as sketched in Fig. 1g, h.

## Atomic force microscopy of the bent Au bottom-electrodes

To quantitatively characterize the surface condition of the Au bottom-electrodes after bending, we performed AFM measurements at a variety of bending radii. Figure 3a–h shows AFM images of the Au surfaces after bending at the ranges considered for the working devices, and we do not observe substantial changes in RMS roughness or any significant fluctuations even at the most convex bending condition (Fig. 3a; RMS = 0.312 nm) and the most concave bending condition (Fig. 3h; RMS = 0.289 nm). Therefore, we conclude that our method of mechanical control does not induce significant change in surface roughness. It has been reported that a rough surface can lead to large amounts of defects in SAMs that results in electrical shorting and large leakage current[31, 32]. The AFM data shown in Fig. 3i, j, framed with the

dashed rectangle, indicates that the Au surfaces can form ruptures at extreme convex $R = -9.4$ mm and humps at extreme concave $R = 9.4$ mm. To avoid any damage induced by extreme-bending-induced changes of surface roughness, we set our maximumly convex bending at $R = -18.3$ mm and maximumly concave bending at $R = 17.0$ mm.

## Sub-nanometer aggregates controlled by mechanical bending

To prove whether we can control the supramolecular aggregates in SAMs by bending the supporting metal substrates, we used fluorescence spectrometry with a 365 nm UV light source to monitor the extent of supramolecular packing at five different bending radii. Figure 4a shows that the peak intensity dramatically increases as the

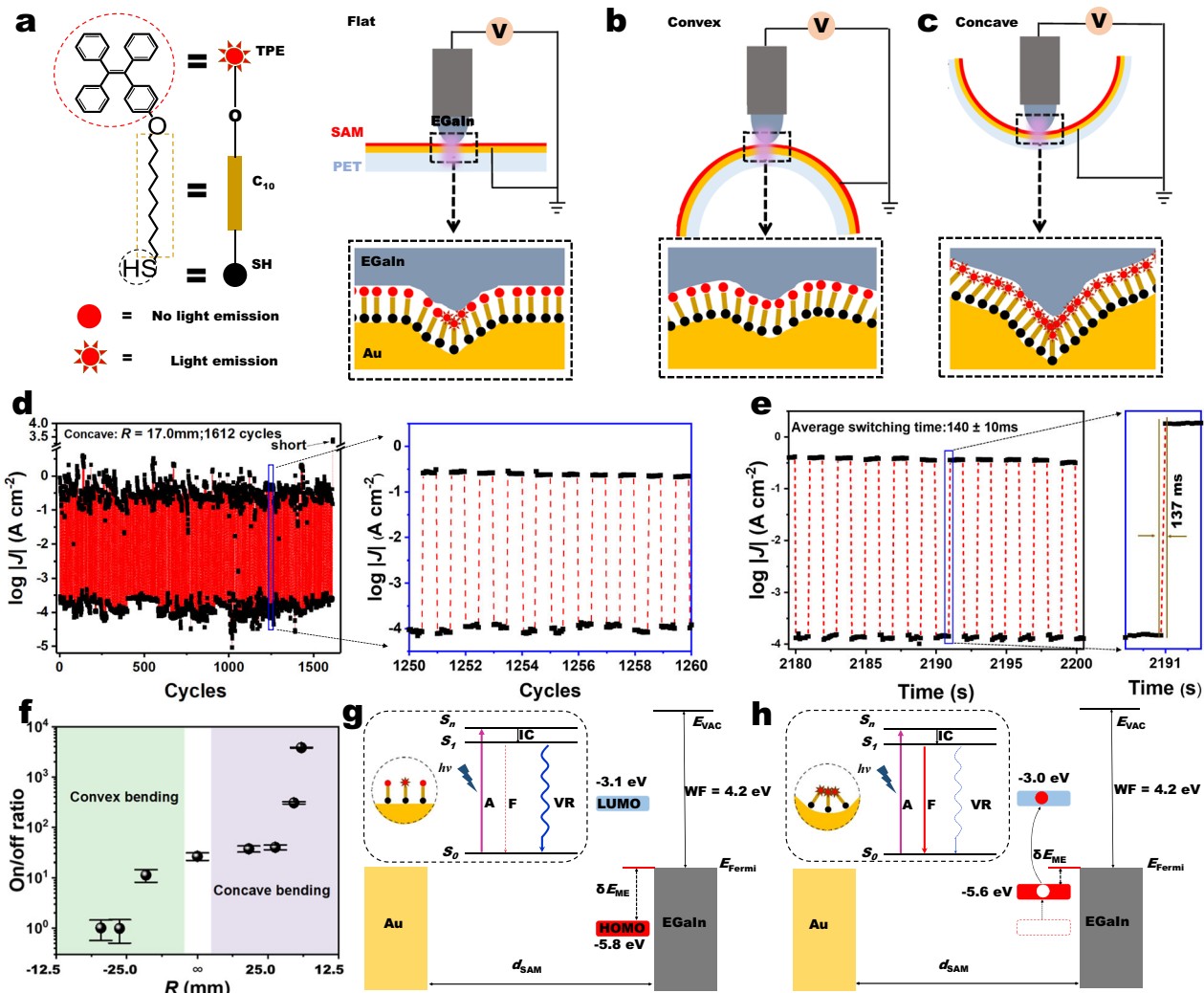

**Fig. 1 | Molecular photoswitching modulated by mechanical bending. a** Left: the molecular structure and corresponding cartoon of the photo-responsive HSC$_{10}$-O-TPE AIE-active molecule. Right: the schematic illustration of the flat PET/Au-SC$_{10}$-O-TPE//Ga$_2$O$_3$/EGaln junction with a UV lamp focused below the junction. The schematic illustration of the **b** convex and **c** concave junctions created by applying opposing forces at two points on the Au top surface or PET back surface. $R$ represents the radius of curvature and is extracted from Supplementary Figs. 59–61. Inside the dashed square, the schematics show the aggregates mainly happen at or near the grain boundaries of Au surfaces. **d** The real-time UV-on/off cycle induced strong switching of log|$J$| at −1.0 V in the concave junction at $R = 17.0$ mm. **e** Zoom-in of sustained switching *vs*. time (the full dataset is in Supplementary Fig. 62) over ten consecutive cycles with UV blinking on and off. The black data points represent the log|$J$| at −1.0 V ($R = 17.0$ mm), and the dashed lines are a guide to the eye. The panel to the right shows one off/on switch cycle and the

average switch time is 140 ± 10 ms (the error bar is the standard deviation obtained from 1115 measurements). **f** The plot of systematic dependence of the on/off ratio on $R$ at −1.0 V (the full dataset is in Supplementary Fig. 65). The energy level diagram of the junctions of Au-SC$_{10}$-O-TPE SAM in **g** flat geometry and **h** concave geometry with UV-on. Two simplified Jablonski diagrams are shown in the insets of panels (**g**) and (**h**). The white dot in the energy level in panel **h** represents the hole while the red dot is the electron. IC internal conversion, A absorption, F fluorescence, VR vibrational relaxation, and $d_{SAM}$ the thickness of the Au-SC$_{10}$-O-TPE SAM. $S_0$ the electronic ground state, $S_1$ the first excited state, $S_n$ the nth excited state, $E_{Fermi}$ the Fermi energy level of EGaln, SOMO singly occupied molecular orbital, HOMO highest occupied molecular orbital, LUMO lowest unoccupied molecular orbital, $\delta E_{ME}$ energy difference between Fermi level and HOMO, $E_{VAC}$ energy level of vacuum, WF work function.

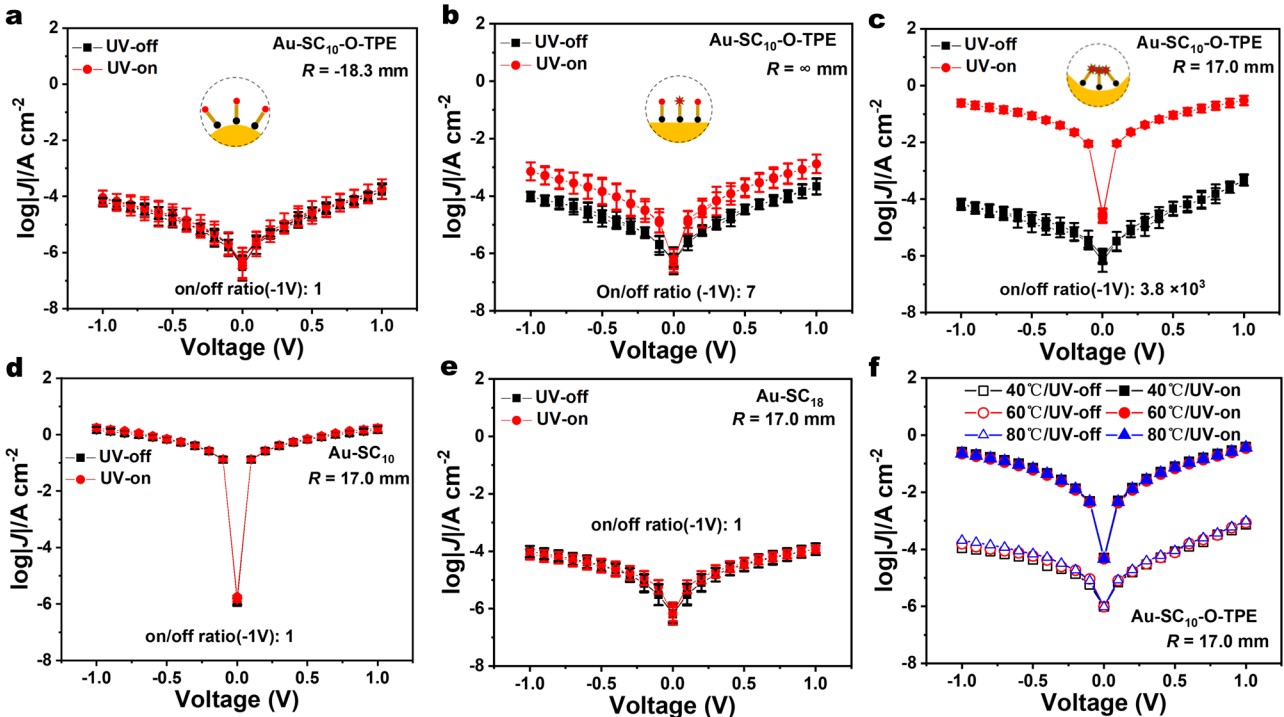

**Fig. 2 | Series of log |J| (V) curves.** The log |J| (V) curves of the Au-SC$_{10}$-O-TPE junction in the UV-off state (black square) and UV-on state (red dot), **a** under convex bending at $R$ = -18.3 mm, **b** flat at $R$ = ∞ mm radii, and **c** concavely bent at $R$ = 17.0 mm. The log |J| (V) values as a function of the voltage of **d** TPE-less Au-SC$_{10}$ junction and **e** TPE-less Au-SC$_{18}$ concave junction. **f** The log |J| (V) curves of the concave Au-SC$_{10}$-O-TPE junction at temperatures of 40, 60, and 80 °C. We followed previously reported procedures to analyze the junction data and Gaussian log-mean values of the J(V) data. All error bars were calculated from the FWHM (full width at half maximum) of the fitted Gaussian distribution curves (Supplementary Figs. 67 and 68).

concavity is increased, which is in good agreement with the reported AIE property of the TPE molecules deduced by comparing behavior in powder and solution[17] Here, we induce the AIE response in the surface-bound monolayers, with the observed dependence on concavity indicating that the mechanically controlled supramolecular aggregation reduces the conformational freedom of the molecules, which damps the rotation and vibrational motion of the bulky TPE headgroups. The bending-induced aggregation artificially locks in a dense π-π network with stronger intermolecular interactions than were naturally available for the sterically unconstrained molecules on the flat, or convex substrate (as substantiated from the packing energy analysis of the molecular dynamics models, Fig. 5 below). Figure 4b maps the change in AIE peak positions as the substrate bends. The peak position shows a small red shift as the concavity is increased, which is in line with AIE measurements of molecules dissolved in poor solvents[33]. Figure 4c illustrates the angular-dependent C K-edge near edge X-ray absorption fine structure (NEXAFS) spectra of Au-SC$_{10}$-O-TPE SAMs at five bending radii, from which we calculated the values of the tilt angle of TPE, $\alpha$, defined as the angle of the π* orbitals located on TPE units against the surface normal (as indicated in Supplementary Fig. 72c; Supplementary Information gives details about the NEXAFS technique). The measured increase of $\alpha$ in the most concave SAMs (Fig. 4d) confirms that the more aggregated Au-SC$_{10}$-O-TPE molecules adopt a distinct, more tilted molecular geometry[34].

To obtain more information regarding the electronic structure of the SAMs as a function of bending, we determined the highest occupied molecular orbital (HOMO) energy of the SAMs by ultraviolet photoelectron spectroscopy (UPS, see all spectra in Supplementary Fig. 72). Figure 4e shows that all the HOMO levels remain far below the Fermi level of the EGaIn top-electrode (-4.2 eV). The HOMO energy shift shown in Fig. 4e does not exhibit any noticeable impact on the conductivity (Fig. 2) or transition voltages (Supplementary Fig. 71); this

is likely due to the Fermi level ($E_F$) pinning[35,36]. The HOMO level of the SAMs is localized at the TPE units (Supplementary Fig. 71a), with the measured small upwards shift tracking the creation of mechanically-constrained strong π-π interactions between TPE units, which is confirmed by the molecular models (Fig. 5, with details in Supplementary Information). Supplementary Table 1 summarizes all the results, and all support our conclusion that we are mechanically directing the aggregation of the AIE-active TPE units in the SAMs to create the high-performance molecular photoswitch. The calculations of the values of HOMO energy level, LUMO energy level, and WF are shown in Supplementary Information, and all values are shown in Supplementary Table 1. All UPS spectra are shown in Supplementary Fig. 72.

**Modeling mechano-controlled AIE in self-assembled monolayers**

To further understand how aggregation at the molecular level in SAMs is affected by the bending of the supporting electrode, molecular dynamics (MD) simulations and time-dependent density functional theory (TD-DFT) calculations were performed. The computed molecular structures in Fig. 5a show that the most concave model (c15 = concave bending angle of 15°, see Supplementary Information theory calculations section on simulations with details in Supplementary Figs. 75 and 76) exhibits the most structuring of the headgroups in the SAM with well-defined short sub-0.5 nm intermolecular distances for π-π interactions (Fig. 5b). The dependence of the computed tilt angle on bending (Supplementary Fig. 72d) is consistent with the trend in the experimentally measured values (Fig. 4d). The highly ordered structure obtained in the most concave junction produces the most stabilizing SAM molecule–molecule packing energy (Fig. 5c), illustrating how improved packing of the adjacent TPE headgroups drives the measured bending-induced spontaneous aggregation.

Quantum mechanical models (TD-DFT, Fig. 5d, full model details are in Supplementary Fig. 75 and Supplementary Information)

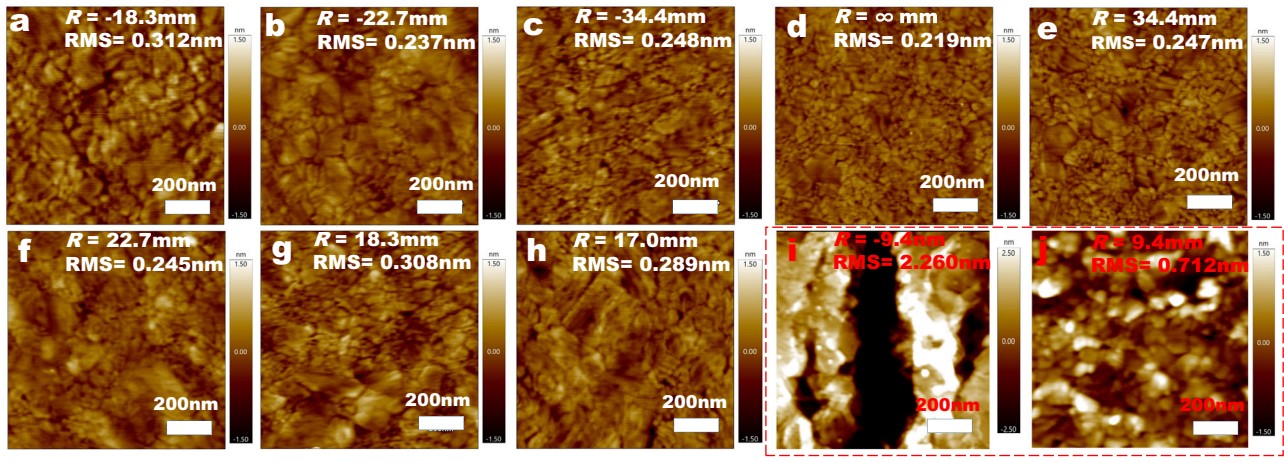

**Fig. 3 | AFM images of the bent Au bottom-electrodes. a** R = -18.3 mm (RMS = 0.312 nm), **b** R = -22.7 mm (RMS = 0.237 nm), **c** R = -34.4 mm (RMS = 0.248 nm), **d** R = ∞ mm (RMS = 0.219 nm), **e** R = 34.4 mm (RMS = 0.247 nm), **f** R = 22.7 mm (RMS = 0.245 nm), **g** R = 18.3 mm (RMS = 0.308 nm), **h** R = 17.0 mm (RMS = 0.289 nm), **i** Control tests of extreme curvature showing significant damage for R = -9.4 mm (RMS = 2.260 nm) and **j** R = 9.4 mm (RMS = 0.712 nm).

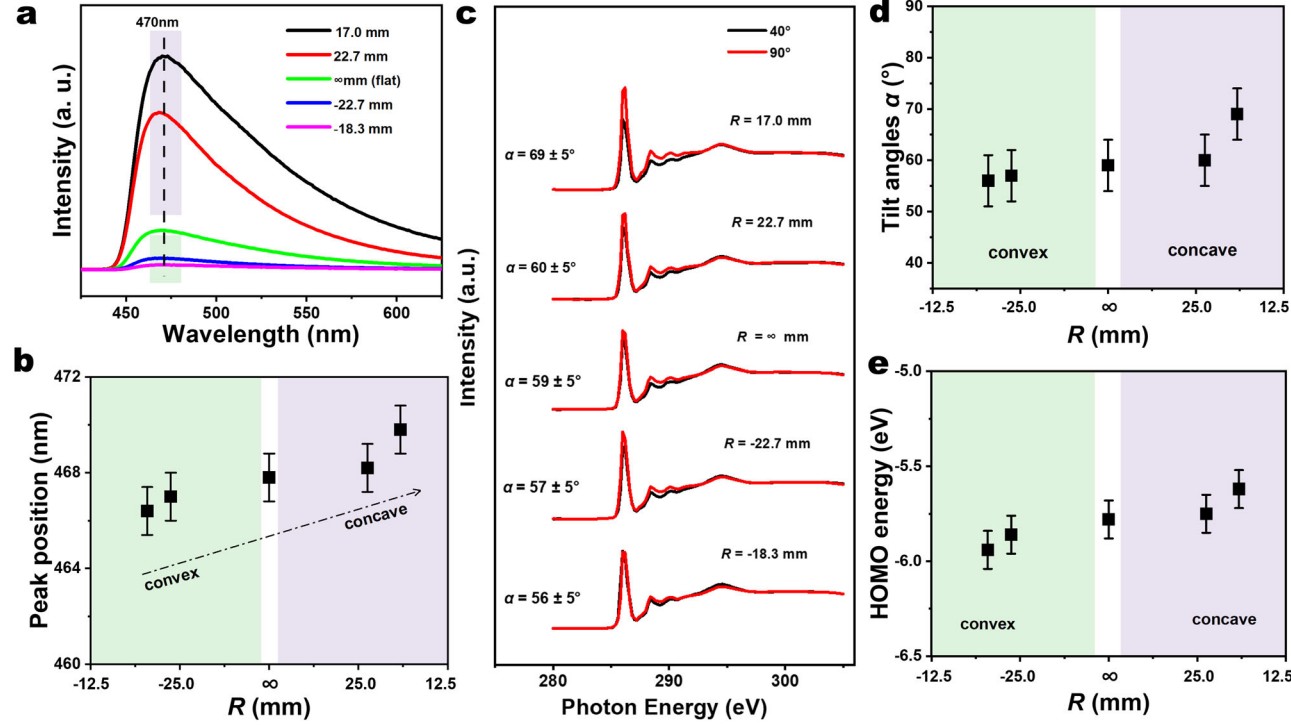

**Fig. 4 | Characterization of the adaptable SAMs by spectroscopy. a** Emission spectrum of Au-$SC_{10}$-O-TPE from convex bending to concave bending under 365 nm exciting light. **b** The upfield shift of the diagnostic peak position near 470 nm in the emission spectrum as a function of R. The error bar represents the systematic error of around 1 nm from the instrument. **c** Angular-dependent C K-edge NEXAFS spectra of Au-$SC_{10}$-O-TPE SAMs acquired at an X-ray incidence angle of 90° (normal incidence, red line) and incidence angle of 40° (grazing incidence, black line) with respect to the surface plane. The definition of α is shown in Supplementary Fig. 72. **d** The tilt angles calculated from angular-dependent C K-edge NEXAFS data as a function of R. The error bar of ±5° includes the angular misalignment due to sample mounting. **e** The plot of measured values of HOMO energy level as a function of R. The error bar is ± 0.1 eV, which comes from the resolution of the spectra.

compared the electronic structure and predicted AIE behavior of the single isolated non-aggregated molecule with various dimer geometries. These include the solid-state 3D single crystal dimer together with representative SAM film dimer geometries (Fig. 5a) found in the most convex and most concave molecular dynamics models (Supplementary Fig. 76). The calculations predict that the dimer found in the crystal maximizes the red shift compared to the single molecule (Fig. 5d), as expected for molecules that are optimally packed without any substrate-imposed steric constraint. The SAM dimer packing geometry in the most concave model produces a well-defined dimer spectrum that closely resembles that of the crystal dimer. By contrast, the computed spectrum from the weaker dimer obtained from the convex models (Supplementary Figs. 77, 78) is closer to the monomer. This shows that curving of the electrode in a concave shape favors a higher ordering of the SAM imposing a 2D crystal-like structure and electronic properties, compared to the more random structure of the flat or convex SAM, supporting the molecular mechanism of bending-induced photoswitching. Our calculations suggest the aggregation of TPE headgroups at concave bending can change the optical property of the SAMs by forming supramolecular aggregates with large

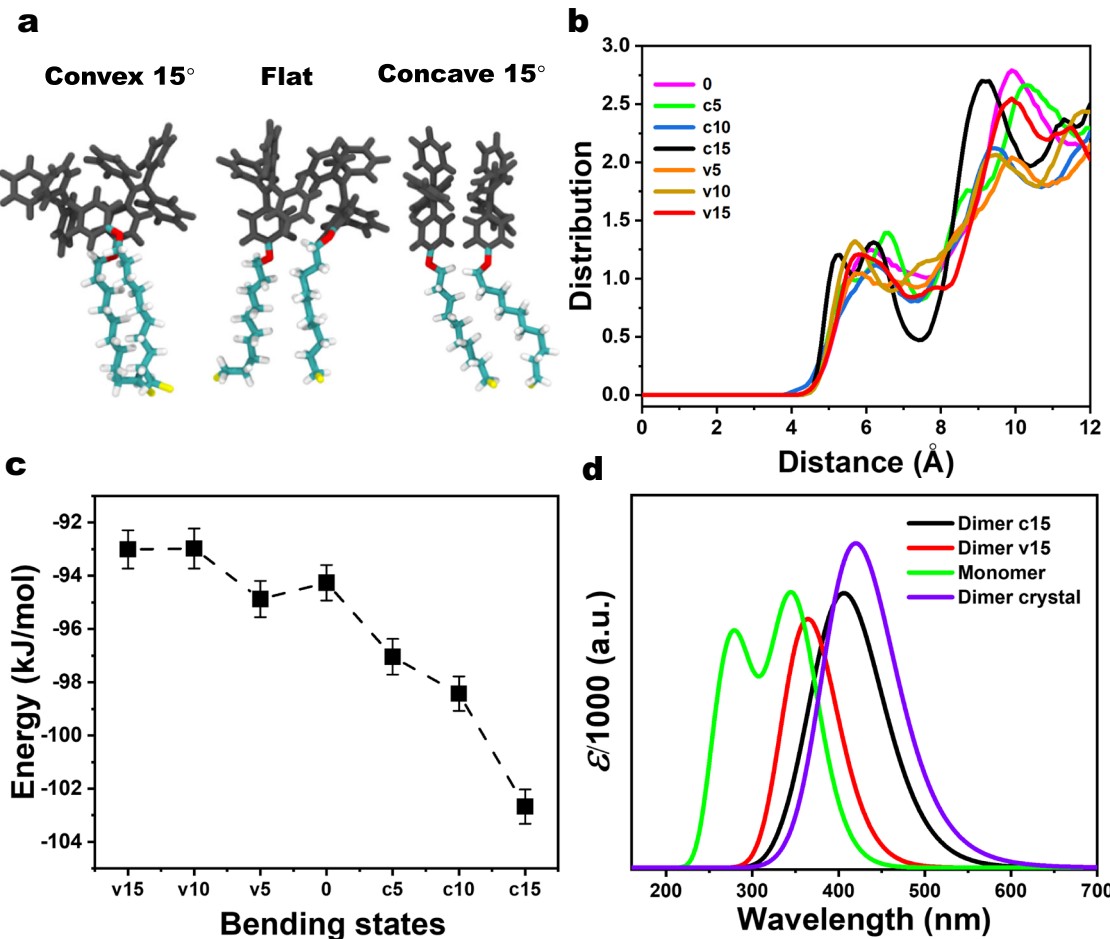

**Fig. 5 | Molecular modeling of the junctions. a** Representative computed structures of Au-SC$_{10}$-O-TPE dimers from molecular dynamics calculations performed with the electrode in (left) convex bending angle of 15°, (middle) flat unbent electrode, and (right) concave bending angle of 15°. Full modeling details and dataset are in methods and Supplementary Information. The alkyl chains are shown as colored sticks (Carbon in cyan, Oxygen in red, Sulfur in yellow, Hydrogen in white), and the TPE headgroups are black. **b** Computed separations between TPE phenyl rings showing the densification of the ordered tight π-π network only in the most concave (c15, concave bending angle of 15°, colored black) SAM structure, in contrast to the looser and more disordered packing in the less concave (c5, c10), flat and convex (v5, v10, v15) geometries. **c** SAM formation energy calculated from the sum of the non-covalent Coulomb and van der Waals supramolecular packing interactions between molecules. The molecule- and time-averaged value is obtained by sampling snapshots of the assembly taken every 100 ps during the final 500 ns of each 1 μs MD simulation. The error bars are calculated from the sum of the sampling snapshots during the taken time. **d** Computed (TD-DFT, B97D/CC-PVTZ) UV-vis spectra of the single isolated AIE-active monomer (green), the crystal dimer (from structure CCDC 973505, purple), the aggregated dimer from the computed most concave SAM structure (c15, black), and the weakly-associated dimer from the computed most convex SAM structure (v15, red). The ε value measures the predicted strength of the electronic transition modulated by a Lorentz function to model thermal effects responsible for thermal broadening of the spectrum. The dimer from the convex junction with minimal π-π stacking (red) shifts towards the monomer, and the aggregated dimer from the ordered, densified π-π network in the concave junction shifts towards the crystal, reflecting the measured shift in the first peak maximum in the spectroscopy experiments shown in Fig. 4b.

molecule–molecule enthalpy and dampened single-molecule entropy, generating an artificially ordered architecture that is not obtainable in flat device geometries.

In our AIE junctions, we measure three orders increase in current density under illumination. We note that large current enhancement under illumination has been observed experimentally by McCreery et al. in junctions with long molecular wires. They proposed hole-electron binding[37,38] as the underlying reason for the enhanced tunneling which was theoretically confirmed for a simple molecular system by Dubi et al.[25]. Therefore, we propose that AIE in our junctions leads to the formation of stable electron-hole pairs that effectively lower the tunneling barriers (Fig. 1g, h). The density of states (DOS) band structure calculations (Supplementary Fig. 79) confirm the lowering of the LUMO energy level close to the Fermi level for the SAMs in the concave junction, which can assist the current enhancement during photoexcitation.

## Molecular engineering to reach on/off ratio of 10⁵

To demonstrate the potential of our rational design strategy of mechano-optoelectronic molecular switching for practical applications in flexible and energy-efficient devices, we synthesized three other AIE-active molecules connected with multiple TPE terminal groups (molecular structures are shown in Fig. 6a, and we name them as di-TPE, tri-TPE, tetra-TPE; see Supplementary Figs. 14–58 for synthesis details). We fabricated photoswitching junctions of these molecules at bending geometry of maximally convex $R = -18.3$ mm, flat $R = \infty$, and maximally concave $R = 17.0$ mm. We observed a steady enhancement of the on/off ratio at $R = 17.0$ mm from $(3.8 \pm 0.1) \times 10^3$ for mono-TPE to $(2.5 \pm 0.2) \times 10^4$ for di-TPE and $(3.3 \pm 0.2) \times 10^5$ for tri-TPE, and plateauing at maximum value of $(4.8 \pm 0.1) \times 10^5$ for tetra-TPE (Fig. 6b–d). To the best of our knowledge, this $10^5$ photoswitching ratio with ~100 ms fast switching is the best performing molecular photoswitch to date (see comparison in Supplementary Table 2). Figure 6d

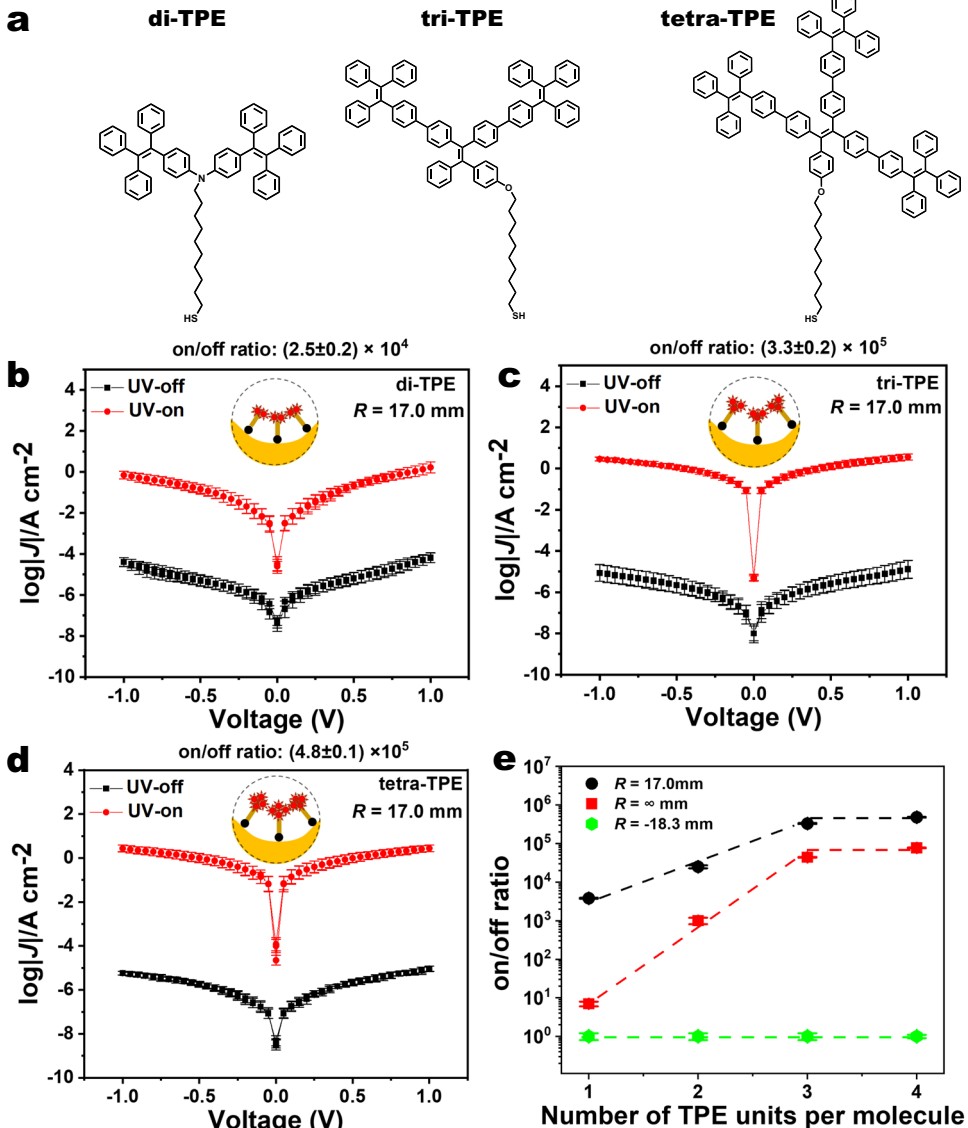

**Fig. 6 | Enhanced photoswitching by molecular engineering. a** Molecular structure of di-TPE, tri-TPE, and tetra-TPE. Optical modulation of log|$J$| across SAMs of **b** di-TPE, **c** tri-TPE, and **d** tetra-TPE. **e** The on/off ratio as a function of number of TPE units per molecule at three different bending geometries: maximumly concave (black), flat (red), and maximumly convex (green). See Supplementary Figs. 68,69 for details including all histograms and fits of $J$ at −1.0 V. The error bars are calculated from Supplementary Figs. 68,69.

shows the exponential increase of on/off ratio when we increase the number of TPE units per molecule from one to three, with an additional mild benefit on maximizing the phenyl density with the very bulky tetra-TPE. Our molecular engineering of photoswitching performance confirms the designability of mechano-optoelectronic response in the junctions, and the ultra-high photoswitching achieved at maximal phenyl density showcases the potential of the devices for unique applications involving coupled mechanical and photonic stimuli/responses.

## Discussion

The fabrication of a high-performance large-area photoswitch controlled by mechanical bending offers a simple and versatile approach for miniaturized devices, based on the self-assembly of AIE-active molecules on a flexible electrode. A large and reproducible on/off ratio of $(3.8 \pm 0.1) \times 10^3$ is obtained by concave bending of the supporting electrode, and the photoswitching is fully reversible for more than 1600 cycles. The on/off ratio can be further enhanced to

$(4.8 \pm 0.1) \times 10^5$ by introducing three more TPEs units in the molecular unit. Our structural characterization supported by atomic-scale models reveals the molecular mechanism of photoswitching via mechanically controlled π-π interactions to efficiently lower the tunneling barrier for charge transport. This work shows the possibility of precisely modulating the function of aggregation-induced emission to realize high-performance electronics with molecular-level control via simple modulation of external mechanical force, which may inspire developments in other fields that also rely on multiple stimuli-responses (light/electricity/mechanics) applied on surfaces or micro-domains, for instance, optomechatronics, tribology, and microfluidic mechanics.

## Methods

### Preparation of the ultra-flat bendable bottom-electrodes

We used template-stripped Au as the ultra-flat surface to grow dense SAMs. The procedure is summarized as follows. We deposited 30-nm-thick Au on clean Si/SiO₂ (100) wafers using a thermal evaporator

(Zhongke Ke Yi). Next, we cut the sheet of polyethylene terephthalate (PET) (thickness of 0.25 mm) into rectangular shape with 75 mm length and ~8 mm width. We used an ultrasonicator (KQ-300DE of Kunshan ultrasonic instrument co. LTD, 40 K Hz) to clean the polyethylene terephthalate (PET) film in 99.9% ethanol for 10 min. The Norland optical adhesive No. 61 was used to glue the PET film on the metal surface, and then solidified at 365 nm UV light for 24 h. We then used a razor blade to peel the Au/PET from the Si/SiO$_2$ wafer before the formation of SAMs.

## Atomic force microscope measurement

The surfaces of the bottom-electrodes at different bending radius was checked by the atomic force microscope (AFM). We recorded the $1 \times 1 \mu m$ AFM data using the Oxford cypher S instrument from Asylum Research at room temperature. We performed the AFM measurements in tapping mode (Cypher VRS, resonant frequency: 150 kHz, force constant: 200 N/m).

## Determination of the radii of electrode-bending

Supplementary Fig. 59 illustrates the process of how we calculate the average radii for both convex and concave bending. First, we captured the images of the bending profiles shown as inserts at each panel of Supplementary Figs. 60 and 61. Second, we used GetData Graph Digitizer software to map the pink color part of the inserts onto 2D plots. Finally, we used a quartic equation to fit the curve, and thus we could calculate the radii ($R$) of the curves by solving the differential of the quartic equation at $x = 0$. In order to distinguish the direction of bending by $R$, we placed a "-" sign before the values of $R$ to represent convexly bent surfaces. Supplementary Figs. 60,61 shows three images around each individual bending radii with their corresponding fits, and the curvature radii with error bars shown in the manuscript were calculated by averaging the results from three individual fits.

## Preparation of self-assembled monolayers

1.0 mM solutions were prepared in distilled ethanol. Then we peeled off the freshly template-stripped PET/Au substrate and immediately immersed it into the solution and kept it in solution for more than 12 h. Finally, the prepared SAM coated substrates were washed with ethanol to remove physisorbed materials followed by gentle drying in a stream of N$_2$.

## Charge transport measurements

We fabricated the SAM-based junctions with cone-shaped tips of Ga$_2$O$_3$/EGaIn following previously reported procedures[39]. Briefly, we biased the EGaIn tips and grounded the bottom-electrodes. For each type of junction, we recorded 20–24 $J(V)$ traces from >20 junctions in voltage range from 0 V → 1.0 V → 0 V → -1.0 V → 0 V. The step size is set at 100 mV and the delay is 0.2 s. The integration time and time resolution of current is 40 ms and 100 ms, respectively. We followed previously reported procedures[40,41] to analyze the junction data and to determine the Gaussian log-mean values of the $J(V)$ data.

## Fluorescence spectroscopy

We used the Edinburgh instruments FLS1000 Fluorescence Spectrometer to record the fluorescence spectra. We measured the emission spectrum under 365 nm excitation light under source light path of Xenon lamp and detector light path of visible PMT-980. The bandwidth of excitation and emission wavelength is 6.0 nm and 17.0 nm, respectively. The step size is 1.0 nm.

## Near edge X-ray absorption fine structure spectroscopy

The NEXAFS spectra at the C K-edge were recorded using the beamlines MCD-A and MCD-B (Soochow Beamline for Energy Materials) at National Synchrotron Radiation Laboratory (NSRLJ) in China. The ultra-high vacuum (UHV) chamber had a base pressure of $1 \times 10^{-10}$ mbar

and measurements are at room temperature. We recorded XPS signals at two different take-off angles for photoelectrons: 90° (normal emission) and 40° (grazing emission). Angular-dependent NEXAFS spectra at C K-edge were recorded in the Auger electron yield (AEY) mode by collecting the Auger electrons resulting from carbon KVV transitions using a Scienta R4000 electron energy analyzer. Linearly $p$-polarized synchrotron light with the degree of linear polarization of 90% was used in the measurements. The photon energy of the incident x-rays was calibrated using a sputter-cleaned gold foil as reference with the photon energy resolution of 200 meV. All spectra were normalized to the incident photon flux monitored by $I_0$ of the refocusing mirror, and then normalized to have the same absorption edge step height well above the absorption edge.

## X-ray and ultraviolet photoelectron spectroscopy

The XPS and UPS were all carried out at the NCESBJ (National Center of Electron Spectroscopy in Beijing). We conducted the XPS spectrum at room temperature in an ultra-high vacuum (UHV) chamber with a base pressure of $1 \times 10^{-8}$ mbar. To probe the valence band, the photon energy at 21.22 eV was used and −10 V bias was applied to the sample to overcome the work function of the analyzer. We used the least-square peak fit analysis with Voigt functions (Lorentzian (30%) and Gaussian (70%)) to fit the XPS spectra with Avantage software, and the sloping background was modeled using Shirley plus linear background correction. All UPS spectra were referenced to the Fermi edge of Au.

## Molecular modeling

For each system, a three-layer slab of (111) gold with surface area 5.19 nm × 5.5 nm is used as the substrate, placed in a unit cell mea u ring 5.19 nm × 5.5 nm × 5.2 nm. 100 molecules are placed in the cell to mimic the estimated experimental surface coverage $\Gamma_{SAM}$ of Au-SC$_{10}$-O-TPE of $5.9 \times 10^{-10}$ mol/cm$^2$ (from XPS, Supplementary Figs. 73 and 74). The TPE headgroups are parametrized with ParamChem[42,43], which provides the CHARMM General Force Field (CGenFF)[44] parameters. Gold parameters are the hydrophilic set of parameters devised for amino acid adsorption by Nawrocki et al.[45]. MD simulations are carried out using the Gromacs 2018.4[46] package with a time step of 2 fs using the Leap frog integrator[47]. Bond lengths to hydrogen are constrained using the LINCS[48] algorithm. Long-range electrostatics are treated by the Particle mesh Ewald (PME) method[49]. The SAM molecule chains and gold atoms are coupled separately to an external heat bath (300 K) with a coupling time constant of 1 ps using the velocity rescaling method[50]. The system is minimized for 100 ps, and equilibrated for 500 ps in the constant volume NVT ensemble. The production runs are split into two phases. Initially the chains are not bound to individual gold atoms on the electrode surface but only constrained to keep their sulfur anchoring groups within bonding distance of the gold surface. This allows the molecules to self-assemble into a uniform film on the surface. This initial physisorption step was carried out for 1 μs, after which each molecule is chemisorbed by bonding its sulfur atom to the closest available gold site. A production phase of 1 μs was then carried out with structures saved every 2 ps. The final structure from the 1 μs simulation is used to prepare the systems with curved substrates. They are placed in a larger box (9.17 nm × 9.17 nm × 9.17 nm) where the gold slab is no longer periodic, to allow the creation of the curved substrates. For reference, a flat, unbent structure was also modeled in this larger simulation cell size. All models were then subjected to a final 1 μs of dynamics with data collected every 2 ps.

The model developed represents a first approximation to the local substrate configuration in the macroscopically bent bottom-electrodes. It is well established that defects away from ideal geometry in the substrate can induce dramatic changes in the SAM conformation and electronic properties[31,51,52]. For all the simulations presented here, the gold atoms are constrained to their starting positions in the flat or bent substrate geometry. Future models could

include local gold substrate restructuring in response to mechanical or optical perturbation, requiring large-scale electronic structure calculations that are becoming feasible with advances in supercomputing[53–55].

All quantum calculations are carried out with Gaussian 16[56], using the B97D[57] functional and the CC-PVTZ basis set. This functional includes van der Waals corrections essential to describe the intermolecular interactions. To reduce computational cost, the tethering $C_{10}$ alkyl chain, which does not participate in the TPE headgroup aggregation, is replaced by a single methyl group. Three dimers are modeled. The first is the dimer taken from the single crystal[58]. The other two are from representative molecular dynamics structures with the electrode curved in the concave or convex geometry.

Finally, transport calculations are performed with DFTB+ software[59]. We consider convex (v15) and concave (v15) dimers in between gold electrodes as depicted in Supplementary Fig. 79. The current, the DOS, and the transmission function are computed at a biasing potential of 0.5 V.

## Data availability
The data supporting the findings of this study are available within the article and its Supplementary Information. The source data are provided with this paper and available in the figshare repository (https://doi.org/10.6084/m9.figshare.24025941). Additional data are available from the corresponding authors on reasonable request.

## Code availability
The code used for the analyses is available from the corresponding authors on reasonable request.

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

## Acknowledgements

Y.L. acknowledges National Natural Science Foundation of China (22273045), and Independent Scientific Research Plan for Young Investigator of Tsinghua University. D.T. acknowledges Science Foundation Ireland (SFI) for support under Grant Number 12/RC/2275_P2 (SSPC) and for supercomputing resources at the SFI/Higher Education Authority Irish Center for High-End Computing (ICHEC).

## Author contributions

D.T. and Y.L. designed the experiments. Z.Y. prepared samples, conducted electrical measurements and performed data analysis. P-A.C. did the theoretical calculations. J-L.L. synthesized and characterized molecules. Z.C. and Z.Y. prepared the bendable bottom-electrodes. N.C., Z.Y. and J-L.L. performed sample characterizations. D.Z., L.D. and C.A.N. helped with data analysis. Z.Y., P-A.C., D.T. and Y.L. contributed mainly to the writing of the manuscript. All authors discussed the results, data analysis, and contributed to the writing of the manuscript.

## Competing interests

The authors declare no competing interests.
