## [Peer Review File · Nature Communications]

REVIEWER COMMENTS

Reviewer #1 (Remarks to the Author):

Yang et al. report photo switching in junctions in which a flexible substrate is used to control aggregation, thus intermolecular interactions that give rise to different degrees of excimer formation, etc., leading to an on/off ratio that varies with the bending radius of the substrate. The introduction frames this work as being faster and more robust than existing reports, but cites a review that includes quite a few examples of switches, some of which appear to be mischaracterized or, at least, conflated. For example, no distinction is made between photoisomerization, which can be non-volatile versus photostationary states, which are not. Characterizing the present work as solving all of the problems of photoswitches at once and therefore being potentially useful for many applications is unnecessary and inaccurate: this work is interesting enough on its own and does not need (borderline disingenuous) puffery.

The comparisons in Table 1 appear to have some errors and omissions. The DAE/BTB switch is reported with a very precise ratio of $(1.0 \pm 0.2) \times 10^4$ and a switching time of 3.6×10^3 , but reference 24 seems only to report the switching ratio as 10 000 and it does not report switching times at all, only that the junctions were illuminated for an hour to maximize the number of switches in the open form. The table also reports the hemicyanine switches as ~ 60 s, but reference 26 reports it as < 60 s, which is ascribed to the instrument resolution, not the switching time. The diarylethene switch is reported as > 1 s but reference 8 states that the persistence time was faster than the 100 ms resolution of the instrument, meaning switching is even faster. Azobenzenes are reported to switch 30 times, but reference 9 shows (from 10.1038/ncomms2937) azobenzenes switching at least 50 times without fatigue. One has to assume that there are more errors in this table.

Scientifically these comparisons (even when made accurately) are illogical. Indeed, the TPE molecules show fast switching, but Figure S31 suggests that it is fast in both directions. Indeed, if the mechanism relies on the presence of molecules in a photo-excited state, they should decay to the ground state on the order of microseconds or faster. That puts them in an entirely different class from many of the switches against which they are compared because they require different inputs to switch between states. Diarylethene, spiropyran, dihydroazulenes, etc. isomerize upon exposure to one wavelength of light and only isomerize back when exposed to a different wavelength (and/or heat). Often they are characterized by their persistence time, as the putative application is the storage or processing of information. The suggestion that TPE switches have applications akin to references 36 and 37 is tacit recognition of this fact (not that molecular electronics has to justify its existence with potential technological applications!) Moreover, molecular switches are rarely observed to switch in operando; most of the reports against which this work is compared switch molecules ex situ and then measure the resulting changes. References 8 and 26 are the most apt comparisons and this work improves on both of those in important ways. Why not just emphasize those comparisons?

In this reviewer's opinion, the interesting aspect of this work is the ability to access molecule-scale phenomenology with macro-scale mechanical perturbation. It is a clever combination of the principles of a break-junction with the ability of monolayer-based junctions to form devices. The fact that the change in aggregation is so clearly visible by UPS, NEXAFS, fluorescence and, most importantly, switching ratios is impressive and sufficiently interesting in its own right not to require an elaborate "our number is bigger than yours" comparison to existing work. If anything, such comparisons distract from the novelty of this work, which is

likely to inspire downstream work that optimizes and leverages the underlying mechano-supramolecular phenomena.

Other considerations:

The UPS results show a clear shift in the HOMO energy as the aggregation changes due to the radius of curvature of the substrate, but the authors are silent as to why this has no discernible impact on the conductivity or transition voltages. Pinning often nullifies substituent effects (particularly in OPEs) so why not mention that fact with some citations?

It is counter-intuitive that the curvatures reported can lead to significant changes in molecular packing in the same way humans do not perceive changes in the curvature of Earth. The calculations used to determine the resulting curvature for the MD simulations are mentioned, but not shown explicitly. It would be nice to see the actual numbers in a table; one would assume that eventually the gold grains would separate or rupture at extreme curvatures and these numbers would help to understand the length scales involved.

An obvious experiment is to grow the SAMs on convex/concave substrates to enhance/suppress the switching when they are bent the other way during measurements. Did the authors try that? If so, did it not have the expected effect?

The switching times are emphasized quite a bit, but the measurement times are hard to follow. The J/V curves are clearly measured slowly and averaged over repeated traces, but the integration time of the current for the real-time data is not reported nor is the time-resolution of the measurement. The dataset in figure 1 d has a lot of noise in it, while the insets look perfectly reproducible; Figure S31 seems to imply that the switching stability (in terms of the values of J at on/off) varies with the radius of curvature; and the stability seems much lower in figure S32 than elsewhere. In addition to disclosing the integration time and time resolution, the variance of J in the on/off states needs to be reported.

Did the authors attempt to measure the current as a function of changing radius of curvature? If not, why? If so, why not show a plot of J in the light/dark as a function of radius of curvature?

Reviewer #2 (Remarks to the Author):

This manuscript reports a device using mechanical bending to control the aggregation of TPE molecules and thus affect their current density. The method is of significance and the results are very interesting. This manuscript is recommended for publication in Nature Communications after careful revisions.

1. Usually, photoswitch means that there is a dynamic structural transformation in photoirradiation or thermal stimulation, and the resulting photocontrol device has the performance of writing and erasing. The UV-controlled on/off switch reported in this manuscript is not the same concept as the molecular photoswitch reported in the literature. The performance in the manuscript depends on the speed of UV blinking on and off. The measured data is not the speed of molecular transformation, but the nature of the excitation light source. It is recommended that the authors read photoswitch literature, modify the manuscript title and carefully revise the text to avoid confusion.
2. AFM is very important device characterization tool. Quantitative analysis results should be given in the text.

3. The authors are advised to carefully check the purity of the product. There exist unassigned peaks in the ^1H NMR spectrum and ^{13}C NMR spectra. Please give pure spectra and assign each ^1H and ^{13}C peaks in the NMR spectra.
4. The intermediates in the synthesis diagram of Supplementary Fig 23 have no synthetic description and characterization.
5. Some recently published papers in the area may be cited and discussed for the benefit of the readers, e.g., *Aggregate* 2023, 4, e245; *Aggregate* 2021, 2, e141.

We express our sincere appreciation to the editor for handling our manuscript and we thank all the expert reviewers for their positive feedback and comments on our work which we have used to prepare the revision. We have fully addressed all reviewer comments and have made the appropriate changes and additions that have improved the quality and usefulness of our paper. A summary of our point-by-point responses is provided below:

Blue italic font: reviewers' comments;

Black font: response to reviewers;

We highlighted all revisions made in the main text and SI in yellow.

REVIEWER COMMENTS

Reviewer #1 (Remarks to the Author):

Yang et al. report photo switching in junctions in which a flexible substrate is used to control aggregation, thus intermolecular interactions that give rise to different degrees of excimer formation, etc., leading to an on/off ratio that varies with the bending radius of the substrate. The introduction frames this work as being faster and more robust than existing reports, but cites a review that includes quite a few examples of switches, some of which appear to be mischaracterized or, at least, conflated. For example, no distinction is made between photoisomerization, which can be non-volatile versus photostationary states, which are not. Characterizing the present work as solving all of the problems of photoswitches at once and therefore being potentially useful for many applications is unnecessary and inaccurate: this work is interesting enough on its own and does not need (borderline disingenuous) puffery.

Response:

We sincerely thank the expert reviewer for finding our work interesting and for their constructive suggestions. We agree with the reviewer that the demonstration of our new molecular switch where we realize mechanical and optical control over the electronic properties is already interesting enough, and we should therefore avoid a reductive mislabelling of our work as one kind of (super)photoswitch. Hence, we have revised our title to “High-Performance Mechano-optoelectronic Molecular Switch”. We also revised the introduction to make sure that the concept of our study is clear and to avoid giving the impression that we are claiming to solve all the problems of photoswitches.

As recommended by the reviewer, we also added a short discussion to clearly differentiate between non-volatile and volatile photoswitches based on either the photoisomerization or photostationary states.

Changes to the manuscript:

On page 1 line 1, we revised the title:

~~Mechanically Controlled High Performance~~ High-Performance Mechano-optoelectronic
Molecular ~~Photoswitch~~ Switch

On page 2 line 20, we revised:

“Here we achieve fully-reversible *in-situ* mechano-optoelectronic switching photoswitching-in self-assembled...”

On page 2 line 27, we revised:

“The best mechano-optoelectronic photoswitching switching occurs...”

On page 3 line 37 and at the section of **References**, we revised:

“...on one switching modality with one stimulus^{4-6 4-8}.”

4 van de Linde, S. & Sauer, M. How to switch a fluorophore: From undesired blinking to controlled photoswitching; *Chem. Soc. Rev.* **43**, 1076-1087 (2014).

5 Bléger, D. & Hecht, S. Visible-light-activated molecular switches. *Angew. Chem. Int. Ed.* **54**, 11338 – 11349 (2015).

6 Huang, X. & Li, T. Recent progress in the development of molecular-scale electronics based on photoswitchable molecules. *J. Mater. Chem. C* **8**, 821-848 (2020).

7 Dulić, D. *et al.* One-way optoelectronic switching of photochromic molecules on gold. *Phys. Rev. Lett.* **91**, 207402 (2003).

8 Jia, C. *et al.* Covalently bonded single-molecule junctions with stable and reversible photoswitched conductivity. *Science* **352**, 1443-1445 (2016).

On page 3 starting on line 39, we changed the following passage from:

“While molecular switches are inherently energy-efficient⁷, theoretically ultrafast molecular photoswitches showed disappointing performance to-date, with small on/off ratio of electric current, poor reproducibility, and slow or stochastic switching^{8,9}. It has been particularly challenging to develop efficient photoswitches in molecular tunnel junctions due to quenching and spontaneous back-switching¹⁰.”

to

“While molecular switches are inherently energy-efficient⁹, theoretically ultrafast molecular photoswitches have shown disappointing performance to-date⁶⁻⁸. Therefore, there is an increasing need to move beyond *ex-situ* switching⁸ and develop efficient *in-situ* molecular switches controlled by light, which has proved challenging due to quenching and spontaneous back-switching^{6,7}.”

On page 3 line 44 we revised:

“...been seldom reported^{10,11}, despite wide implementation of mechanically-controlled switches¹¹⁻¹³ and recent report of miniaturized opto-electro-mechanical switching using a 25-nm gold membrane¹⁴. Furthermore, the range of applications in fields such as circularly polarized luminescence (CPL) has been expanded using stimuli of temperature, humidity, and pH¹⁵. While *ex-situ* photoswitching has found some applications, such as in gold nanocluster-based fluorescence photoswitching¹⁶, further advances require *in-situ* and multi-stimuli switching techniques. Here, we use....”

- 9 Raymo, F. M. Digital Processing and communication with molecular switches. *Adv. Mater.* **14**, 401-414 (2002).
- 10 Berson, J., Moosmann, M., Walheim, S. & Schimmel, T. Mechanically induced switching of molecular layers. *Nano Lett.* **19**, 816-822 (2019).
- 11 Frisenda, R., Janssen, V. A. E. C., Grozema, F. C., van der Zant, H. S. J. & Renaud, N. Mechanically controlled quantum interference in individual π -stacked dimers. *Nat. Chem.* **8**, 1099-1104 (2016).
- 12 Jeong, H. Y. *et al.* Graphene oxide thin films for flexible nonvolatile memory applications. *Nano Lett.* **10**, 4381-4386 (2010).
- 13 Dai, S. *et al.* Intrinsically ionic conductive cellulose nanopapers applied as all solid dielectrics for low voltage organic transistors. *Nat. Commun.* **9**, 2737 (2018).
- 14 Haffner, C. *et al.* Nano-opto-electro-mechanical switches operated at CMOS-level voltages. *Science.* **366**, 860-864 (2019).
- 15 He, Y., Lin, S., Guo, J. & Li, Q. Circularly polarized luminescent self-organized helical superstructures: From materials and stimulus-responsiveness to applications. *Aggregate.* **2**, e141 (2021).
- 16 Zhong, W., Yan, X., Qu, S. & Li, Shang. Site-specific fabrication of gold nanocluster-based fluorescence photoswitch enabled by the dual roles of albumin proteins. *Aggregate.* **4**, e245 (2023).

On page 4 line 61, we replaced the original ref 17 with a more appropriate reference for molecular photoswitches:

“...molecular photoswitches^{6,17} rely on optically triggered conformational changes such as photoisomerization¹⁹⁻²⁰ ...or photocyclization²¹⁻²³. However...”

6 Huang, X. & Li, T. Recent progress in the development of molecular-scale electronics based on photoswitchable molecules. *J. Mater. Chem. C* **8**, 821-848 (2020).

The other references given above are:

- 19 Smaali, K. *et al.* High on–off conductance switching ratio in optically-driven self-assembled conjugated molecular systems. *ACS Nano* **4**, 2411-2421 (2010).
- 20 Comstock, M. J. *et al.* Reversible photomechanical switching of individual engineered molecules at a metallic surface. *Phys. Rev. Lett.* **99**, 038301 (2007).
- 21 Hnid, I., Frath, D., Lajoie, F., Sun, X. & Lacroix, J.-C. Highly efficient photoswitch in diarylethene-based molecular junctions. *J. Am. Chem. Soc.* **142**, 7732-7736 (2020).
- 22 Roldan, D. *et al.* Charge transport in photoswitchable dimethyldihydropyrene-type single-molecule junctions. *J. Am. Chem. Soc.* **135**, 5974-5977 (2013).
- 23 Li, T. *et al.* Ultrathin reduced graphene oxide films as transparent top-contacts for light switchable solid-state molecular junctions. *Adv. Mater.* **25**, 4164-4170 (2013).

On page 4 line 65, we added:

“...Those photoswitching processes are non-volatile and have promising molecular device applications in information storage and processing^{6,8,12,15}. However, these non-volatile photoswitches cannot provide the *in-situ* fast switching required by memory devices in electrical circuits^{12,19}. In contrast to the photoisomerization-based approach, photo-assisted conduction via photostationary states is a volatile switch. ~~On the other hand~~ For example...”

The comparisons in Table 1 appear to have some errors and omissions. The DAE/BTB switch is reported with a very precise ratio of $(1.0 \pm 0.2) \times 10^4$ and a switching time of 3.6×10^3 , but reference 24 seems only to report the switching ratio as 10 000 and it does not report switching times at all, only that the junctions were illuminated for an hour to maximize the number of switches in the open form. The table also reports the hemicyanine switches as ~60 s, but reference 26 reports it as < 60 s, which is ascribed to the instrument resolution, not the switching time. The diarylethene switch is reported as > 1 s but reference 8 states that the persistence time was faster than the 100 ms resolution of the instrument, meaning switching is even faster. Azobenzenes are reported to switch 30 times, but reference 9 shows (from 10.1038/ncomms2937) azobenzenes switching at least 50 times without fatigue. One has to assume that there are more errors in this table. Scientifically these comparisons (even when made accurately) are illogical. Indeed, the TPE molecules show fast switching, but Figure S31 suggests that it is fast in both directions. Indeed, if the mechanism relies on the presence of molecules in a photo-excited state, they should decay to the ground state on the order of microseconds or faster. That puts them in an entirely different class from many of the switches against which they are compared because they require different inputs to switch between states.

Response:

We express our gratitude for the reviewer's valuable suggestions. The reviewer is correct that it is unfair to compare non-volatile and volatile photoswitches together in a table with the same parameters. Therefore, we removed Table 1. We also carefully reviewed and doublechecked all the references and performance measures cited in the manuscript. As described further below, we now include a concise comparison table, supplementary Table 2, in the Supplementary Information (SI) to provide a clear overview.

Changes to the manuscript:

On page 4 line 73, we deleted:

“To date, the best photoswitches (see summary of molecular photoswitches in Table 1) were demonstrated...”

On page 4 line 73, we revised:

“...from low on/off ratios ($\sim 1.4 \ll 20$) and low...”

On page 4 line 75, we revised:

“...with on/off ratio of **more than** 10^2 and switching time **of less than 100 m~1s**. Chiechi *et. al.*²⁶ showed large-area self-assembled monolayers (SAMs) of hemicyanine molecules generating a similar on/off ratio of ~ 100 with switching time of **less than** 60 s. The challenge then to achieve simultaneous large on/off ratio, fast, stable, reversible ***in-situ*** switching in...”

Diarylethene, spiropyran, dihydroazulenes, etc. isomerize upon exposure to one wavelength of light and only isomerize back when exposed to a different wavelength (and/or heat). Often they are characterized by their persistence time, as the putative application is the storage or processing of information. The suggestion that TPE switches have applications akin to references 36 and 37 is tacit recognition of this fact (not that molecular electronics has to justify its existence with potential technological applications!)

Response:

We appreciate the referee's comment, and we fully agree with their perspective. In accordance with the suggestion, we have removed the specific examples of potential technological applications.

Changes to the manuscript:

On page 18 line 351, we deleted:

“for instance flexible antennas and optical controlled nano-robot/mechatronic sensors^{36,37}.”

and edited to read:

“...of the devices for unique applications involving coupled mechanical and photonic stimuli/responses.”

Moreover, molecular switches are rarely observed to switch in operando; most of the reports against which this work is compared switch molecules ex situ and then measure the resulting changes. References 8 and 26 are the most apt comparisons and this work improves on both of those in important ways. Why not just emphasize those comparisons?

Response:

We thank the reviewer for the suggestion. Following the reviewer’s recommendation, we removed Table 1 and included a concise comparison table, Supplementary Table 2, in the Supplementary Information (SI) to provide a clear overview.

Change to the manuscript:

On page 18 line 345, we revised:

“... (see comparison in Supplementary Table 2 Table 1)”

Change to the supplementary information:

On page S70, we added Supplementary Table 2:

Supplementary Table 2. Comparison of *in-situ* molecular switching

Photoswitching units	On/off ratio	Switch time	On/off cycles	Ref.
Diarylethene	$(1.1 \pm 0.6) \times 10^2$	<100 ms	$>1.0 \times 10^2$	S11
Hemicyanine	<100	<60 s	50	S12
mono-TPE	$(3.8 \pm 0.1) \times 10^3$	140 ± 10 ms	1.6×10^3	This work
tetra-TPE	$(4.8 \pm 0.1) \times 10^5$	143 ± 12 ms	1.2×10^3	This work

In this reviewer's opinion, the interesting aspect of this work is the ability to access molecule-scale phenomenology with macro-scale mechanical perturbation. It is a clever combination of the principles of a break-junction with the ability of monolayer-based junctions to form devices. The fact that the change in aggregation is so clearly visible by UPS, NEXAFS, fluorescence and, most importantly, switching ratios is impressive and sufficiently interesting in its own right not to require an elaborate "our number is bigger than yours" comparison to existing work. If anything, such comparisons distract from the novelty of this work, which is likely to inspire downstream work that optimizes and leverages the underlying mechano-supramolecular phenomena.

Response:

We very much appreciate the positive and constructive feedback. In accordance with the reviewer's suggestion, we have removed Table 1 along with the "our number is bigger than yours" related sentences from the manuscript. We also considerably altered the Introduction as shown above.

Other considerations:

The UPS results show a clear shift in the HOMO energy as the aggregation changes due to the radius of curvature of the substrate, but the authors are silent as to why this has no discernible impact on the conductivity or transition voltages. Pinning often nullifies substituent effects (particularly in OPEs) so why not mention that fact with some citations?

Response:

We thank the reviewer for pointing out the shift of HOMO energy. The reviewer is correct that we should mention this observation and refer to the corresponding literature on Fermi level pinning.

Change to the manuscript:

On page 13 line 243 we revised:

“...far below the Fermi level of the EGaIn top-electrode (-4.2 eV). The HOMO energy shift shown in Fig. 4e does not exhibit any noticeable impact on the conductivity (Figure 2) or transition voltages (Supplementary Fig. 71); this is likely due to the Fermi level (E_F) pinning^{36,37}. The HOMO level of the SAMs is localized at the TPE units (Supplementary Fig. 71a), with the...”

36 Braun, S.; Salaneck, W. R.; Fahlman, M. Energy-level alignment at organic/metal and organic/organic interfaces. *Adv. Mater.* **21**, 1450–1472(2009)

37 Cahen, D.; Kahn, A.; Umbach, E. Energetics of molecular interfaces. *Mater. Today* **8**, 32–41(2005)

Change to the supplementary information:

On page S58, we added Supplementary Fig. 71:

Supplementary Fig 71. a, $\text{Log}|J|$ as a function of curvature in the dark; Inset is the computed TPE-SAM molecule HOMO surface (see DFT methods section for details). **b-d**. Fowler–Nordheim (FN) plots for Au-SC₁₀-O-TPE, derived from the current-density data in Fig. 2a-c at different bending radii as shown.

It is counter-intuitive that the curvatures reported can lead to significant changes in molecular packing in the same way humans do not perceive changes in the curvature of Earth. The calculations used to determine the resulting curvature for the MD simulations are mentioned, but not shown explicitly. It would be nice to see the actual numbers in a table; one would

assume that eventually the gold grains would separate or rupture at extreme curvatures and these numbers would help to understand the length scales involved.

Response:

We thank the reviewer for raising this excellent point. It is indeed something we were aware of and considered in developing the model. As shown in Supplementary Fig. 61, we can estimate the angle corresponding to the circular part of the curve.

Supplementary Fig. 61. The real bending curves (inset) and fitted mathematic curves (red line) at various concave bending angles. We measured four different extents of concave bending and each R was fitted three times. Supplementary Fig. 59d shows the average R results. The concave $R = 34.4 \pm 0.3$ mm was fitted from **a**, **b** and **c**. $R = 22.7 \pm 0.6$ mm. was fitted from **d**, **e** and **f**. $R = 18.3 \pm 0.1$ mm was fitted from **g**, **h** and **i**. $R = 17.0 \pm 0.1$ mm was fitted from **j**, **k** and **l**.

At most, the circular part corresponds to the x-values between -10 and 10 mm. The arc is obviously longer than 20 mm, but to a first approximation, this gives an angle of 32° from Supplementary Fig. 61a. To determine the radius angles, we drew the blue circle, which had a calculated radius of 34.3mm, to align with the bending curves depicted in Figure R1 with the angle of about 32° .

Fig. R1. The method used to estimate the bending angles.

Applying the same approach to the size of the gold slab used for the molecular SAM model gives a trivially small angle of $8.7e-6^\circ$. Yet, experimentally, the curvature affects the switching response. To rationalise this, we consider that the experimental SAM substrate is not perfect but contains grains, step edges and other defects and imperfections. These “weak spots” would be locally more sensitive to any change in the curvature than an ideal flat surface. Both the geometric and electronic structure is affected as the band gap is populated by isolated energy levels that work as steps and facilitate the transport. The wavefunction can then propagate to the rest of the SAM thanks to the supramolecular correlation, that is either enhanced or weakened by the curvature of the device. Our curved MD models bent at an angle of $5-15^\circ$ can be considered as an extreme case of such defects where external conditions are amplified compared to the rest of the SAM. Please note that we tested larger angles of 30° and 45° but these produced large repulsions between the molecules and physically unrealistic changes in the packing structures of the SAM. As we use a purely classical model, we must constrain the gold atoms throughout the simulations to their starting positions in order to preserve the convex or concave geometry. This is further underlined and discussed in the revised manuscript, including the requested details of the calculations used to determine the resulting curvature for the MD simulations and table of numbers used.

Changes to the manuscript:

From page 27, line 572, to page 28, line 579, we added:

The model developed represents a first approximation to the local substrate configuration in the macroscopically bent bottom-electrodes. It is well established that defects away from ideal geometry in the substrate can induce dramatic changes in the SAM conformation and electronic properties^{32,51-52}. For all the simulations presented here, the gold atoms are constrained to their starting positions in the flat or bent substrate geometry. Future models could include local gold substrate restructuring in response to mechanical or optical perturbation, requiring large-scale electronic structure calculations that are becoming feasible with advances in supercomputing⁵³⁻⁵⁵.

32 Nijhuis, C. A., Reus, W. F. & Whitesides, G. M. Mechanism of rectification in tunneling junctions based on molecules with asymmetric potential drops. *J. Am. Chem. Soc.* **132**, 18386-18401 (2010).

...

51 Gannon, G. *et al.* Molecular dynamics study of naturally occurring defects in self-assembled monolayer formation. *ACS Nano* **4**, 921-932 (2010).

52 Yuan, L. *et al.* On the Remarkable role of surface topography of the bottom electrodes in blocking leakage currents in molecular diodes. *J. Am. Chem. Soc.* **136**, 6554-6557 (2014).

53 Thompson, D. *et al.* Formation mechanism of metal–molecule–metal junctions: molecule-assisted migration on metal defects. *J. Phys. Chem. C.* **119**, 19438-19451 (2015).

54 Griffiths, J. *et al.* Resolving sub-angstrom ambient motion through reconstruction from vibrational spectra. *Nat. Commun.* **12**: 6759 (2021).

55 Carnegie, C. *et al.* Flickering nanometre-scale disorder in a crystal lattice tracked by plasmonic flare light emission. *Nat. Commun.* **11**, 682 (2020).

Change to the Supplementary Information:

From page S64, line 588 of Supporting Information, we added:

To create the curved SAM models, the length l_x of the gold substrate along the bending axis, here the x-axis, is divided by the chosen angle θ . This gives the base radius of curvature r_c :

$$r_c = \frac{l_x}{\theta}$$

The system is shifted so that the z-position of the bottom layer of gold atoms is set at zero, and the middle of the xy plane coincides with the center of the gold slab. For each atom i in the model, the effective radius of curvature r_i is defined as follows:

$$r_i = r_c + z_i$$

The angle formed between the atom and the center of curvature is:

$$\theta_i = \frac{x_i}{r_i}$$

The new position of the atom is then:

$$\begin{cases} x'_i = r_i \sin \theta_i \\ z'_i = r_i \cos \theta_i - r_c \end{cases}$$

Supplementary Fig. 76 shows the curved systems, both convex and concave, for three angles of curvature: 5, 10, 15°, and Table S3 shows the corresponding radius of curvature for each model.

Supplementary Fig. 76. Computed SAM structures from MD simulations with the supporting gold electrode bent at 5°, 10°, 15° in a convex fashion (**a, b, c, g, h, i**) and in a concave fashion (**d, e, f, g, k, l**). The starting structures are shown in **a-f**. The final structures are shown in **g-l**. The gold atoms are shown as van der Waals spheres, the SAM alkyl chains as sticks (Carbon in cyan, Oxygen in red, Sulfur in yellow, Hydrogen in white), and the AIE-active headgroups as grey sticks.

On page S70, we added Supplementary Table 3:

Supplementary Table 3. Radius of curvature for MD models

Angle of Curvature [°]	Radius of Curvature [nm]
5	92.53
10	46.26
15	30.84

From the experimental observations, the reviewer is correct that the gold film will eventually separate or rupture at extreme curvatures. We added Fig. 3i and 3j in the main text to show AFM images of gold substrates bent at extreme $R = -9.4$ mm and $R = 9.4$ mm. We observed rupture at convex $R = -9.4$ mm and hump formation at concave $R = 9.4$ mm. We chose the less extreme maximal convex ($R = -18.3$ mm) and concave ($R = 17.0$ mm) bending radii based on these AFM results, to produce bent but not damaged substrates for our device measurements.

Change to the manuscript:

On page 10 line 193, we added:

“Atomic force microscopy (AFM) of the bent Au bottom-electrodes. To quantitatively characterize the surface condition of the Au bottom-electrodes after bending, we performed AFM measurements at a variety of bending radii. Fig. 3a-h shows AFM images of the Au surfaces after bending at the ranges considered for the working devices, and we do not observe substantial changes in RMS roughness or any significant fluctuations even at the most convex bending condition (Fig. 3a; RMS = 0.312 nm) and the most concave bending condition (Fig. 3h; RMS = 0.289 nm). Therefore, we conclude that our method of mechanical control does not induce significant change in surface roughness. It has been reported that a rough surface can lead to large amounts of defects in SAMs that results in electrical shorting and large leakage current³²⁻³³. The AFM data shown in Fig. 3i and 3j, framed with the dashed rectangle, indicates that the Au surfaces can form ruptures at extreme convex $R = -9.4$ mm and humps at extreme concave $R = 9.4$ mm. To avoid any damage induced by extreme-bending-induced changes of surface roughness, we set our maximumly convex bending at $R = -18.3$ mm and maximumly concave bending at $R = 17.0$ mm.”

Fig. 3. AFM images of the bent Au bottom-electrodes. a, $R = -18.3$ mm (RMS = 0.312 nm), b, $R = -22.7$ mm (RMS = 0.237 nm), c, $R = -34.4$ mm (RMS = 0.248 nm), d, $R = \infty$ mm (RMS = 0.219 nm), e, $R = 34.4$ mm (RMS = 0.247 nm), f, $R = 22.7$ mm (RMS = 0.245 nm), g, $R = 18.3$ mm (RMS = 0.308 nm), h, $R = 17.0$ mm (RMS = 0.289 nm), i, Control tests of extreme curvature showing significant damage for $R = -9.4$ mm (RMS = 2.260 nm) and j, $R = 9.4$ mm (RMS = 0.712 nm)

32 Nijhuis, C. A., Reus, W. F. & Whitesides, G. M. Mechanism of rectification in tunneling junctions based on molecules with asymmetric potential drops. *J. Am. Chem. Soc.* **132**, 18386-18401 (2010).

33 Jiang, L., Sangeeth, C. S. S., Yuan, L., Thompson, D. & Nijhuis, C. A. One-nanometer thin monolayers remove the deleterious effect of substrate defects in molecular tunnel junctions. *Nano Lett.* **15**, 6643–6649(2015)

An obvious experiment is to grow the SAMs on convex/concave substrates to enhance/suppress the switching when they are bent the other way during measurements. Did the authors try that? If so, did it not have the expected effect?

Response:

We thank the referee for this excellent suggestion. We were happy to follow the recommendation to grow SAMs on convex/concave substrates and subsequently bend and measure them in the opposite direction. We performed the following two additional experiments:

1. Grow SAMs on concavely bent substrates, and then bend and measure them in convex geometry (supplementary Fig. 70a and b);
2. Grow SAMs on convexly bent substrates, and then bend and measure them in concave geometry (supplementary Fig. 70c and d);

Our measurements (supplementary Fig. 70a-d) show that the $J(V)$ response is independent of the initial substrate curvature. Thus, the conformation of the substrate that the SAM is grown on does not make a difference. Only the subsequent bending of the SAM-coated substrate matters.

Change to the Supplementary Information:

On page S57, we added Supplementary Fig. 70:

Supplementary Fig. 70. SAMs grown on concavely bent substrates, and then bent and measured in convex geometry: **a**, The logarithm of the absolute value of the current density ($\log|J|$) versus the voltage (V), and **b**, the corresponding histograms of $\log|J|$ at -1.0 V. SAMs grown on convexly bent substrates, and then bent and measured in concave geometry: **c**, The logarithm of the absolute value of the current density ($\log|J|$) versus the voltage (V), and **d**, the corresponding histograms of $\log|J|$ at -1.0 V.

The switching times are emphasized quite a bit, but the measurement times are hard to follow. The J/V curves are clearly measured slowly and averaged over repeated traces, but the integration time of the current for the real-time data is not reported nor is the time-resolution of the measurement. The dataset in figure 1 d has a lot of noise in it, while the insets look perfectly reproducible; Figure S31 seems to imply that the switching stability (in terms of the values of J at on/off) varies with the radius of curvature; and the stability seems much lower in figure S32 than elsewhere. In addition to disclosing the integration time and time resolution, the variance of J_{in} the on/off states needs to be reported.

Response:

We thank the referee for this comment. As suggested, we reported the integration time and time resolution of the current in the revised Methods section. They are 40 ms and 100 ms, respectively. We added the resolution of the measurement in Supplementary Fig. 64. We agree with the reviewer that Fig 1d shows all current density over the full 1612 cycles and our zoom-in is a relatively less noisy segment consisting of 10 cycles. We now show additional zoom-in

areas in the response figure below (Supplementary Fig. 63). The variance of J in the on/off states is shown in Supplementary Fig. 62 and 65.

The stability of J at Supplementary Fig. 32 (now Supplementary Fig. 65) varies with the radius of curvature. We also found that the values of J at small bending curvatures respond more weakly to UV light, and the junctions, in some cycles, did not switch by light (Supplementary Fig. 65 panel e). We consider the reason could be i) the weak aggregation at small bending curvature and the dynamic variation of molecules causes an unstable current response during the consecutive $J(V)$ measurements; ii) the vibration from our junction setup. In response to the referee comment, we performed the statistical analysis of the switch response to curvature.

Changes to manuscript:

On page 25 line 514, we added:

“...the delay is 0.2 s. The integration time and time resolution of current is 40 ms and 100 ms, respectively. We followed...”

Changes to Supplementary Information:

On page S50, we added Supplementary Fig. 62:

Supplementary Fig. 62. The real-time on-off cycles of Au-SC₁₀-O-TPE at $R = 17.0$ mm with **a**, 1115 and **b**, 1096 cycles. **c**, The real-time on-off cycles of Au-SC₁₀-O-tetraTPE at $R = 17.0$ mm with 1150 cycles. **d**, Zoom-in of sustained switching vs. time (the full dataset is in panel c) over ten consecutive cycles with UV blinking on and off. The black data points represent the $\log |J|$ at -1.0 V ($R = 17.0$ mm), and the dashed lines are a guide to the eye. The average switching time is 143 ± 13 ms obtained from the >1000 cycles in panel c. **e**, Corresponding histogram of the data in panel c. **f**, Histogram of the data in Fig. 1d. **g**, Histogram of the data in panel a. **h**, Histogram of the data in panel b.

On page S51, we added Supplementary Fig. 63:

Supplementary Fig. 63. a, The real-time on-off cycles of Au-SC₁₀-O-TPE at $R = 17.0$ mm during 1612 cycles. Zoom-in of b, ~100 and c, ~195 sustained switching cycles.

On page S51, we added Supplementary Fig. 64:

Supplementary Fig. 64. a, The real-time on-off switching of Au-SC₁₀-O-TPE at $R = 17.0$ mm. b, The real-time on-off switching of Au-SC₁₀-O-tetraTPE at $R = 17.0$ mm. c, The histogram of data in panel a. d, The histogram of data in panel b.

On page S52, we added:

Supplementary Fig. 65. The *in-situ* on/off switch cycle of Au-SC₁₀-O-TPE SAM at bending radii of **a**, -18.3 mm, **c**, -22.7 mm, **e**, -34.4 mm, **g**, flat, **i**, 34.4 mm, **k**, 22.7 mm, **m**, 18.3 mm and **o** 17.0 mm. The corresponding histograms for **a**, **c**, **e**, **g**, **i**, **k**, **m**, and **o**, are plotted in panels **b**, **d**, **f**, **h**, **j**, **l**, **n**, and **p**, respectively.

Did the authors attempt to measure the current as a function of changing radius of curvature? If not, why? If so, why not show a plot of J in the light/dark as a function of radius of curvature?

Response:

We thank the referee for the very constructive advice. We attempted the suggested experiments, but unfortunately, we found that measurement of current density as a function of continuously changed radius is impossible using our current junction structures (Fig. 1a-1c). The reason is that when we attempted to bend the bottom electrode in operando, the EGaIn tips did not readily follow the vertical movement of the samples. To address this issue, we are currently starting to develop a transparent and fully bendable molecular device (Fig. R2) that may allow us perform the suggested experiment. We will present the results in a follow-up paper in the context of creating a fully soft and highly integrated molecular device.

Fig R2. **a**, The image of the flexible and transparent nature of the cross-bar molecular device. **b**, An optical microscopy image of the 10×10 crossbar array. **c**, Photographs of the device with a 10×10 crossbar array of molecular junctions after the transfer of patterned EGaIn top-electrodes. **d**, A schematic representation of the 10×10 molecular junction array.

Reviewer #2 (Remarks to the Author):

This manuscript reports a device using mechanical bending to control the aggregation of TPE molecules and thus affect their current density. The method is of significance and the results are very interesting. This manuscript is recommended for publication in Nature Communications after careful revisions.

Response:

We thank the reviewer for finding our work significant and interesting, and for their recommendation for publication following careful revisions. As described below, we used the expert, constructive comments of the referee to carefully revise and strengthen the study.

1. Usually, photoswitch means that there is a dynamic structural transformation in photoirradiation or thermal stimulation, and the resulting photocontrol device has the performance of writing and erasing. The UV-controlled on/off switch reported in this manuscript is not the same concept as the molecular photoswitch reported in the literature. The performance in the manuscript depends on the speed of UV blinking on and off. The measured data is not the speed of molecular transformation, but the nature of the excitation light source. It is recommended that the authors read photoswitch literature, modify the manuscript title and carefully revise the text to avoid confusion.

Response:

We thank the referee sincerely for bringing this to our attention. We apologize for the confusion caused by our nomenclature. Following your suggestion, we have simplified the title, which now reads "High-performance mechano-optoelectronic molecular switch." Furthermore, we have carefully and thoroughly studied all the literature on phototswitches and made important changes to the text to ensure clarity and prevent any further confusion. Because this comment is similar to the first comment of reviewer #1, please also refer to our response to that comment above.

Changes to manuscript:

On page 1 line 1, we revised the title:

~~Mechanically Controlled High-Performance Molecular Photoswitch~~ High-Performance Mechano-optoelectronic Molecular Switch

On page 2 line 20, we revised:

“Here we achieve fully-reversible *in-situ* mechano-optoelectronic switching photoswitching in self-assembled...”

On page 2 line 27, we revised:

“The best mechano-optoelectronic photoswitching switching occurs...”

On page 3 line 37 and at the section of **References**, we revised:

“...on one switching modality with one stimulus^{4-6,8}...”

“... ”

4 van de Linde, S. & Sauer, M. How to switch a fluorophore: From undesired blinking to controlled photoswitching; *Chem. Soc. Rev.* **43**, 1076-1087 (2014).

5 Bléger, D. & Hecht, S. Visible-light-activated molecular switches. *Angew. Chem. Int. Ed.* **54**, 11338 – 11349 (2015).

6 Huang, X. & Li, T. Recent progress in the development of molecular-scale electronics based on photoswitchable molecules. *J. Mater. Chem. C* **8**, 821-848 (2020).

7 Dulić, D. *et al.* One-way optoelectronic switching of photochromic molecules on gold. *Phys. Rev. Lett.* **91**, 207402 (2003).

8 Jia, C. *et al.* Covalently bonded single-molecule junctions with stable and reversible photoswitched conductivity. *Science* **352**, 1443-1445 (2016).

...”

On page 3 line 39, we changed the following sentences from:

“While molecular switches are inherently energy-efficient⁷, theoretically ultrafast molecular photoswitches showed disappointing performance to-date, with small on/off ratio of electric current, poor reproducibility, and slow or stochastic switching^{8,9}. It has been particularly challenging to develop efficient photoswitches in molecular tunnel junctions due to quenching and spontaneous back-switching¹⁰.”

to

“While molecular switches are inherently energy-efficient⁹, theoretically ultrafast molecular photoswitches have shown disappointing performance to-date⁶⁻⁸. Therefore, there is an increasing need to move beyond *ex-situ* switching⁸ and develop efficient *in-situ* molecular

switches controlled by light, which has proved challenging due to quenching and spontaneous back-switching^{6,7}..”

On page 3 line 44, we revised:

“...been seldom reported¹⁰⁺¹¹, despite wide implementation of mechanically-controlled switches¹¹⁻¹³ and recent report of miniaturized opto-electro-mechanical switching using a 25-nm gold membrane¹⁴. Furthermore, the range of applications in fields such as circularly polarized luminescence (CPL) has been expanded using stimuli of temperature, humidity, and pH¹⁵. While *ex-situ* photoswitching has found some applications, such as in gold nanocluster-based fluorescence photoswitching¹⁶, further advances require *in-situ* and multi-stimuli switching techniques. Here, we use....”

On page 4 line 61, the original ref 17 is not the best reference for “molecular photoswitches”, and therefore we replaced the reference:

“...molecular photoswitches^{6,17} rely on optically triggered conformational changes such as photoisomerization¹⁹⁻²⁰...or photocyclization²¹⁻²³. However...”

6 Huang, X. & Li, T. Recent progress in the development of molecular-scale electronics based on photoswitchable molecules. *J. Mater. Chem. C* **8**, 821-848 (2020).

On page 4 line 65, we added:

“...Those photoswitching processes are non-volatile and have promising molecular device applications in information storage and processing^{6,8,12,15}. However, these non-volatile photoswitches cannot provide the *in-situ* fast switching required by memory devices in electrical circuits^{12,19}. In contrast to the photoisomerization-based approach, photo-assisted conduction via photostationary states is a volatile switch. ~~On the other hand~~ For example,...”

The references are given as follows:

- 19 Smaali, K. *et al.* High on–off conductance switching ratio in optically-driven self-assembled conjugated molecular systems. *ACS Nano* **4**, 2411-2421 (2010).
- 20 Comstock, M. J. *et al.* Reversible photomechanical switching of individual engineered molecules at a metallic surface. *Phys. Rev. Lett.* **99**, 038301 (2007).
- 21 Hnid, I., Frath, D., Lafalet, F., Sun, X. & Lacroix, J.-C. Highly efficient photoswitch in diarylethene-based molecular junctions. *J. Am. Chem. Soc.* **142**, 7732-7736 (2020).
- 22 Roldan, D. *et al.* Charge transport in photoswitchable dimethyldihydropyrene-type single-molecule junctions. *J. Am. Chem. Soc.* **135**, 5974-5977 (2013).
- 23 Li, T. *et al.* Ultrathin reduced graphene oxide films as transparent top-contacts for light switchable solid-state molecular junctions. *Adv. Mater.* **25**, 4164-4170 (2013).

Changes to the Supplementary Information:

On page S1 line 1, we revised the title:

~~Mechanically Controlled High Performance~~ High Performance Mechano-optoelectronic
~~Molecular Photoswitch~~

2. AFM is very important device characterization tool. Quantitative analysis results should be given in the text.

Response:

We thank the referee for the comment. We agree with the reviewer that the AFM is very important and should be provided in the main text. As suggested, we moved AFM pictures from the Supplementary information to Fig 3 and provided analysis on page 10.

Changes to manuscript:

On page 10, we added:

“Atomic force microscopy (AFM) of the bent Au bottom-electrodes. To quantitatively characterize the surface condition of the Au bottom-electrodes after bending, we performed AFM measurements at a variety of bending radii. Fig. 3a-h shows AFM images of the Au surfaces after bending at the ranges considered for the working devices, and we do not observe substantial changes in RMS roughness or any significant fluctuations even at the most convex bending condition (Fig. 3a; RMS = 0.312 nm) and the most concave bending condition (Fig. 3h; RMS = 0.289 nm). Therefore, we conclude that our method of mechanical control does not induce significant change in surface roughness. It has been reported that a rough surface can lead to large amounts of defects in SAMs that results in electrical shorting and large leakage current³²⁻³³. The AFM data shown in Fig. 3i and 3j, framed with the dashed rectangle, indicates that the Au surfaces can form ruptures at extreme convex $R = -9.4$ mm and humps at extreme concave $R = 9.4$ mm. To avoid any damage induced by extreme-bending-induced changes of surface roughness, we set our maximumly convex bending at $R = -18.3$ mm and maximumly concave bending at $R = 17.0$ mm.”

On page 11, we added Fig. 3:

Fig. 3. AFM images of the bent Au bottom-electrodes. a, $R = -18.3$ mm (RMS = 0.312 nm), b, $R = -22.7$ mm (RMS = 0.237 nm), c, $R = -34.4$ mm (RMS = 0.248 nm), d, $R = \infty$ mm (RMS = 0.219 nm), e, $R = 34.4$ mm (RMS = 0.247 nm), f, $R = 22.7$ mm (RMS = 0.245 nm), g, $R = 18.3$ mm (RMS = 0.308 nm), h, $R = 17.0$ mm (RMS = 0.289 nm), i, Control tests of extreme curvature showing significant damage for $R = -9.4$ mm (RMS = 2.260 nm) and j, $R = 9.4$ mm (RMS = 0.712 nm).

3. The authors are advised to carefully check the purity of the product. There exist unassigned peaks in the ^1H NMR spectrum and ^{13}C NMR spectra. Please give pure spectra and assign each ^1H and ^{13}C peaks in the NMR spectra.

Response:

We have added the NMR spectra of all synthesized compounds and assigned each peak. To save the length of this response letter, please see the revised Supplementary Fig 2 to Supplementary Fig 58 in the Supplementary Information.

4. The intermediates in the synthesis diagram of Supplementary Fig 23 have no synthetic description and characterization.

Response:

We thank the referee for catching this oversight and we have added the synthetic description and characterization of all the intermediates as shown on page S3 to page S45.

5. Some recently published papers in the area may be cited and discussed for the benefit of the readers, e.g., Aggregate 2023, 4, e245; Aggregate 2021, 2, e141.

Response:

We thank the referee for the suggestions and we added the citations.

Changes to the manuscript:

On page 3 line 47, we added:

“...Furthermore, the range of applications in fields such as circularly polarized luminescence (CPL) has been expanded using stimuli of temperature, humidity, and pH¹⁵. While *ex-situ* photoswitching has found some applications, such as in gold nanocluster-based fluorescence photoswitching¹⁶, further advances require *in-situ* and multi-stimuli switching techniques. Here...”

15 He, Y., Lin, S., Guo, J. & Li, Q. Circularly polarized luminescent self-organized helical superstructures: From materials and stimulus-responsiveness to applications. *Aggregate*. **2**, e141 (2021).

16 Zhong, W., Yan, X., Qu, S., & Li, Shang. Site-specific fabrication of gold nanocluster-based fluorescence photoswitch enabled by the dual roles of albumin proteins. *Aggregate*. **4**, e245 (2023).

REVIEWERS' COMMENTS

Reviewer #1 (Remarks to the Author):

The main shortcomings of the initial submission were non-scientific. The citations and comparisons with existing literature conflated key features of switching and molecular switches to the point that it was unclear what was being claimed in the present work. The authors have remedied those mistakes and the revised manuscript is much clearer about what the specific claims are and how they compare to existing work.

The relatively minor scientific issues have been addressed with additional experiments and more detailed descriptions of methodology that clarify the underlying mechanism of switching and its limitations and characteristics. The explanation of how the changing curvature translates into significant changes to intermolecular distances, in particular, is a big improvement. The hypothesis that the different interactions are driven by molecules at or near grain boundaries makes intuitive sense and can (in future work) be tested experimentally. The manuscript can be published in its current form, but the authors should consider adding one last figure showing this hypothesis schematically, to make it clear that the cartoon insets in, for example, Figs 1 and 2 are just cartoons summarizing a concept and that the actual effect is driven as much by the topography of the substrate as it is by curvature.

Reviewer #2 (Remarks to the Author):

The authors have adequately revised their manuscript according to my previous comments and suggestions. The quality of the manuscript has been improved after the revision. The revised manuscript is recommended for publication in its present form.

REVIEWER COMMENTS

Reviewer #1 (Remarks to the Author):

The main shortcomings of the initial submission were non-scientific. The citations and comparisons with existing literature conflated key features of switching and molecular switches to the point that it was unclear what was being claimed in the present work. The authors have remedied those mistakes and the revised manuscript is much clearer about what the specific claims are and how they compare to existing work.

The relatively minor scientific issues have been addressed with additional experiments and more detailed descriptions of methodology that clarify the underlying mechanism of switching and its limitations and characteristics. The explanation of how the changing curvature translates into significant changes to intermolecular distances, in particular, is a big improvement. The hypothesis that the different interactions are driven by molecules at or near grain boundaries makes intuitive sense and can (in future work) be tested experimentally. The manuscript can be published in its current form, but the authors should consider adding one last figure showing this hypothesis schematically, to make it clear that the cartoon insets in, for example, Figs 1 and 2 are just cartoons summarizing a concept and that the actual effect is driven as much by the topography of the substrate as it is by curvature.

Response:

We sincerely thank the reviewer for constructive suggestions. As suggested, we added panels in Fig.1.

Changes to manuscript:

We revised the Fig.1:

Fig. 1. Molecular photoswitching modulated by mechanical bending. **a**, Left: the molecular structure and corresponding cartoon of the photo-responsive HSC₁₀-O-TPE AIE-active molecule. Right: the schematic illustration of the flat PET/Au-SC₁₀-O-TPE//Ga₂O₃/EGaIn junction with a UV lamp focused below the junction. The schematic illustration of the **b** convex and **c** concave junctions created by applying opposing forces at two points on the Au top surface or PET back surface. R represents the radius of curvature and is extracted from Supplementary Fig. 59-61. **Inside the dashed square, the schematics show the aggregates mainly happen at or near the grain boundaries of Au surfaces.** **d**, The real-time UV-on/off cycle induced strong switching of $\log|J|$ at -1.0 V in the concave junction at $R = 17.0$ mm. The total number of cycles is 1612 and the panel to the right shows the zoom-in between 1250 to 1260 cycles. **e**, Zoom-in of sustained switching vs. time (the full dataset is in Supplementary Fig. 62) over ten consecutive cycles with UV blinking on and off. The black data points represent the $\log|J|$ at -1.0 V ($R = 17.0$ mm), and the dashed lines are a guide to the eye. The panel to the right shows one off/on switch cycle and the average switch time is 140 ± 10 ms (the error bar is the standard deviation obtained from 1115 measurements). **f**, The plot of systematic dependence of the on/off ratio on R at -1.0 V (the full dataset is in Supplementary Fig. 65). The energy level diagram of the junctions of Au-SC₁₀-O-TPE SAM in **g** flat geometry and **h** concave geometry with UV-on. Two simplified Jablonski diagrams are shown in the insets of panels **g** and **h**. The white dot in the energy level in panel **h** represents the hole while the red dot is the electron. IC: internal conversion, A: absorption, F: fluorescence, VR: vibrational relaxation, and d_{SAM} : the thickness of the Au-SC₁₀-O-TPE SAM. S_0 : the electronic ground state; S_1 : the first excited state; S_n : the n^{th} excited state; E_{Fermi} : the Fermi energy level of EGaIn; SOMO: singly occupied molecular orbital; HOMO: highest occupied molecular orbital;

LUMO: lowest unoccupied molecular orbital; δE_{ME} : energy difference between Fermi level and HOMO; E_{VAC} : energy level of vacuum; WF: work function.

Reviewer #2 (Remarks to the Author):

The authors have adequately revised their manuscript according to my previous comments and suggestions. The quality of the manuscript has been improved after the revision. The revised manuscript is recommended for publication in its present form.

Response:

We sincerely thank the reviewer for constructive suggestions.